

# Glacial-interglacial contrasts in the marine inorganic carbon chemistry of the Benguela Upwelling System

Szabina Karancz[1]; Lennart J. de Nooijer[1]; Bas van der Wagt[1]; Marcel T. J. van der Meer[2], Sambuddha Misra[3]; Rick Hennekam[1]; Zeynep Erdem[2]; Julie Lattaud[4]; Negar Haghipour[5,6];

Stefan Schouten[2,7]; Gert-Jan Reichart[1,7]

[1] Department of Ocean Systems, NIOZ Royal Netherlands Institute for Sea Research, Texel, The Netherlands

[2] Department of Marine Microbiology and Biogeochemistry, NIOZ Royal Netherlands Institute for Sea Research, Texel, The Netherlands

[3] Centre for Earth Sciences, Indian Institute of Science, Bangalore, India

[4] Department of Environmental Sciences, University of Basel, Basel, Switzerland

[5] Geological Institute, Department of Earth Sciences, ETH Zürich, Zürich, Switzerland

[6] Laboratory of Ion Beam Physics, ETH Zürich, Zürich, Switzerland

[7] Department of Earth Sciences, Faculty of Geosciences, Utrecht University, Utrecht, The Netherlands

*Correspondence to: Szabina Karancz (szabina.karancz@nioz.nl)*





**Abstract.** Upwelling regions are dynamic systems where relatively cold, nutrient- and $CO_2$-rich waters reach to the surface from the deep. $CO_2$ sink or source properties of these regions are dependent not only on the dissolved inorganic carbon content of the upwelled waters, but also on the efficiency of the biological carbon pump that provides constraint on the drawdown of $pCO_2$ in the surface waters. The Benguela Upwelling System (BUS) is a major upwelling region with one of the most productive marine ecosystems today. However, contrasting signals reported on the variation in upwelling intensities based on, for instance, foraminiferal and radiolarian indices from this region over the last glacial cycle indicate that a complete understanding of (local) changes is currently lacking. To reconstruct changes in the $CO_2$ history of the Northern Benguela upwelling region over the last 27 ka BP, we used a box core (64PE450-BC6) and piston core (64PE450-PC8) from the Walvis Ridge. Here, we apply various temperature and $pCO_2$-proxies, representing both surface ($U^{K'}_{37}$, $\delta^{13}C$ of alkenones) and intermediate depth (Mg/Ca, B/Ca, S/Mg, $\delta^{11}B$ in planktonic foraminiferal shells) processes. Reconstructed $pCO_2$ records suggest enhanced storage of carbon at depth during the last glacial maximum. The offset between $\delta^{13}C$ of planktonic (high $\delta^{13}C$) and benthic foraminifera (low $\delta^{13}C$) suggests an evidence of a more efficient biological carbon pump, potentially fuelled by remote and local iron supply through aeolian transport and dissolution in the shelf regions, effectively preventing release of the stored glacial $CO_2$.

## 1 Introduction

Upwelling systems are crucial components in the global carbon cycle thanks to intense biogeochemical cycling and enhanced biological productivity (Turi et al., 2014). Upwelling zones return the cold, nutrient- and $CO_2$-rich waters from depth to the surface which is also reflected in regional changes in surface water inorganic carbon chemistry. The connection between the deep and surface ocean thereby provides a potential mechanism linking changes in ocean circulation and chemistry with the atmosphere. Still, the shoaling of the thermocline and nutricline in these regions also favours phytoplankton growth to such a degree that these areas represent majority of the most productive regions of the ocean (Fig. 1 a). Thus, the leakage of $CO_2$ from the depths to the atmosphere is negated by biological sequestration, simultaneously rendering its quantification a challenging undertaking. The surface waters of the upwelling system undergo an increase in the partial pressure of $CO_2$ ($pCO_2$) and decrease in pH due to upwelling of deep $CO_2$-rich water. In turn, the enhanced primary productivity due to increased nutrients result in drawdown of $pCO_2$ by converting $CO_2$ into organic carbon, after which it may be returned to the deep ocean via the biological carbon pump (BCP; Volk and Hoffert, 1985; Longhurst and Glen Harrison, 1989; Ducklow et al., 2001; Turi et al., 2014; Hales et al., 2005; Muller-Karger et al., 2005). Ultimately, the net $CO_2$ flux from the ocean to the atmosphere is a function of the balance between upwelling strength (increase in $CO_2$) and efficiency of the BCP (drawdown of $CO_2$). On geological time scales this efficiency may have varied, potentially modulating glacial-interglacial cycling of $CO_2$ (Kohfeld et al., 2005; Kwon et al., 2009; Parekh et al., 2006; Hain et al., 2014).

The efficiency of the BCP determines how much of newly produced particulate organic carbon at the surface is transported to the deep (Volk and Hoffert, 1985; Hain et al., 2014). During primary production, nutrients are consumed (e.g., nitrate, phosphate; Redfield, 1958) from the surface ocean and dissolved inorganic carbon (DIC) is taken up in organic matter, which is also reflected by the enrichment in $^{13}C$ of the surface DIC (Degens et al.,



1968). This implies that we can use seawater carbon isotopes as proxy for the efficiency of the BCP. Seawater

carbon isotopes can be reconstructed using the carbon isotopic composition (δ[13]C) in shells of carbonate

producers, such as foraminifera. During high productivity periods the enhanced carbon uptake at the sea surface

will enrich the shells of planktonic foraminifera in [13]C. At the same time, the [13]C-depleted carbon transported to

the deep as organic matter will decrease the [13]C content of the deep water DIC pool, resulting in low δ[13]C values

in the benthic foraminifera shells (Fig. 1 b). Therefore, the difference between planktonic and benthic δ[13]C

provides a measure for the efficiency of the BCP, where more divergent values indicate a more efficient BCP

(Hilting et al., 2008).

Reconstruction of inorganic carbon chemistry can be used to constrain past changes in $CO_2$-flux between the

ocean and atmosphere (Fig. 1 c). Reconstruction of the complete inorganic carbon system is based on at least two

parameters of this system ($pCO_2$, $[CO_3^{2-}]$, $[HCO_3^-]$, pH, [DIC] and total alkalinity), as well as on the knowledge

of temperature and salinity (Zeebe and Wolf-Gladrow, 2001). Commonly used tracers for constraining parameters

of the marine inorganic carbon chemistry are based on both organic (e.g., δ[13]C of alkenones; Pagani et al., 2002;

Pagani, 2014; Popp et al., 1998; Laws et al., 1995) and inorganic (e.g., δ[11]B of foraminifera shells; Hemming and

Hanson, 1992; Palmer and Pearson, 2003; Foster and Rae, 2016) proxy signal carriers, although these proxies

rarely agree completely for upwelling regions (Seki et al., 2010; Palmer et al., 2010) or in general (Rae et al.,

2021).

Here we compare organic and inorganic proxies for temperature ($U^{K'}_{37}$, Mg/Ca) and the carbon system (alkenone-

δ[13]C, foraminiferal-δ[11]B) with reconstructed efficiency of the BCP in the Benguela upwelling area to unravel the

potential role of such areas in the known changes in atmospheric $pCO_2$ on glacial-interglacial time scales. Proxies

for seawater carbon chemistry have specific inherent complications and their application require critical

assumptions. Therefore, we here also explore the recently suggested S/Mg ratio as a proxy for $[CO_3^{2-}]$ (Karancz

et al., 2024a) as an additional constraint for past seawater carbon chemistry.

We focus on the Benguela Upwelling System (BUS) as it is one of the major upwelling regions, where strength

of the upwelling and productivity changed over glacial timescales. Whether upwelling intensity was stronger

during glacial periods (Oberhänsli, 1991; Little et al., 1997; Kirst et al., 1999; Mollenhauer et al., 2003) or

interglacial periods (Diester-Haass et al., 1992; Des Combes and Abelmann, 2007) is, however, still debated.

Inconsistencies in the published body of work is possibly caused by seasonal differences between proxy signal

carriers and/or major spatial (depth) related gradients, which is especially true for regions with strong $CO_2$ flux

dynamics. Exchange of $CO_2$ between seawater and atmosphere at these regions may be constrained only by

applying multiple proxies that comprise various living depth and seasonal preference. This at the same time allows

comparing proxies and investigate (in)consistencies between different carbon system- and temperature proxies.





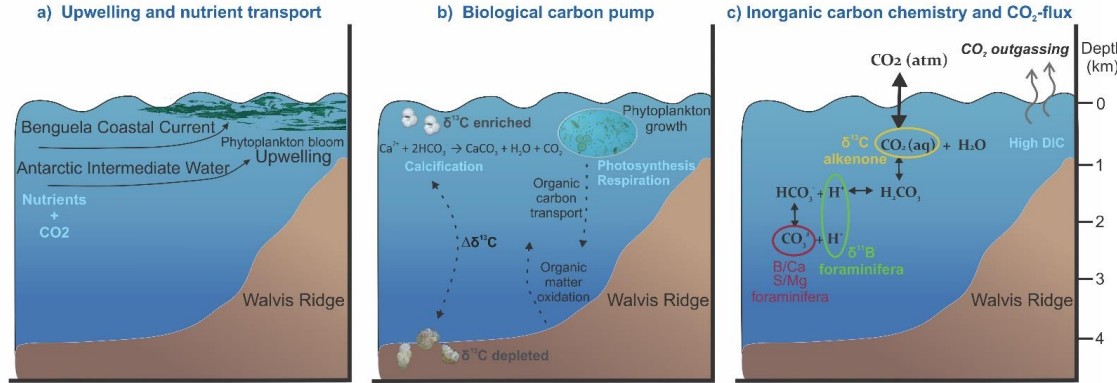

**Figure 1: Cross sections of the Benguela Upwelling System depicting the characteristics of an upwelling region, where a) nutrient- and CO₂-rich waters are upwelled to the surface, b) high productivity contributes to the drawdown of CO₂ in the surface layers via the biological carbon pump, and c) the upwelling strength and efficiency of the biological carbon determines variations in the marine inorganic carbon chemistry, and hence CO₂-flux.**

## 2 Oceanographic setting

The BUS is one of the four major Eastern Boundary Upwelling Systems, it is located between 15° and 34° S along the coastline of Africa (Hill, 1998; Hart and Currie, 1960). This region bears the highest productivity today among the Eastern Boundary Upwelling Systems, fuelled by nutrients transported mainly from the higher latitudes. Advection of the cold and nutrient rich water is a persistent phenomenon throughout the year (Carr, 2001; Chavez and Messié, 2009) and the magnitude of the particulate organic carbon (POC) flux from the surface to the deep exceeds 20 gC m$^{-2}$ yr$^{-1}$ (Henson et al., 2011; Laws et al., 2000; Devries and Weber, 2017).

The BUS is associated with the South Atlantic anticyclonic gyre which gives rise to upwelling on its' southeastern flank where it meets the African continent (Peterson and Stramma, 1991). The low-pressure system over western South Africa causes a pressure gradient between the continent and the ocean and thereby strengthen the southerly wind stress off the coast of Angola and Namibia. The interplay between the equatorward trade winds, the Coriolis-force, and the presence of the continental boundary lead to the offshore transport of surface waters. As such, this causes coastal upwelling of nutrient-rich South Atlantic Central Water (formed in the western South Atlantic; Stramma and England, 1999) and Antarctic Intermediate Water (AAIW). The upwelled waters are transported equatorward along the coast of Africa via the Benguela Current (BC) giving rise to high biological productivity. Filaments of high productive waters can be seen extending from the African continent (Fig. 2). Finally, the Walvis Ridge potentially plays a role in affecting local hydrography and hence the position of the upwelling (Peterson and Stramma, 1991).

The BC with its two main branches, the Benguela Oceanic Current and the Benguela Coastal Current, is the major northward flowing component of the BUS which joins the poleward flowing Angola Current in the north



(Stramma and England, 1999). This convergence zone is located between 15°S and 18°S and is known as the

Angola-Benguela frontal zone. The upwelling zone is bounded by warm current systems, the Angola Current

system in the north and the Agulhas Current system in the south (Shannon and Nelson, 1996; Shillington, 1998;

Shannon and O'toole, 2003). Hence, the BC is composed of a mixture of waters originated not only from the mid-

latitude surface waters of the Central Southern Atlantic Ocean and the Southern Ocean but also from the Indian

Ocean (Gordon, 1986; Lutjeharms and Valentine, 1987). This creates a north-to-south decrease in surface water

temperature and salinity in the region (Santana-Casiano et al., 2009). Hydrographic changes in the region over

glacial cycles have been related to changes in the transfer of Indian Ocean waters through Agulhas leakage

variability (Knorr and Lohmann, 2003; Peeters et al., 2004; Scussolini and Peeters, 2013).

The BUS region is characterized by year-round upwelling but intensity varies over time due to the seasonal shift

of the South Atlantic gyre. This results in stronger upwelling intensities in June-August compared to the rest of

the year (Santana-Casiano et al., 2009; Kämpf and Chapman, 2016). The spatial and temporal dynamics of the

BUS result in large variability in the associated $CO_2$ flux. Predominantly, it acts as a $CO_2$ source (Laruelle et al.,

2014; Brady et al., 2019; Roobaert et al., 2019), but this may be interrupted by periods during which it acts as a

$CO_2$ sink due to the high primary productivity (Gruber et al., 2009; Gregor and Monteiro, 2013).


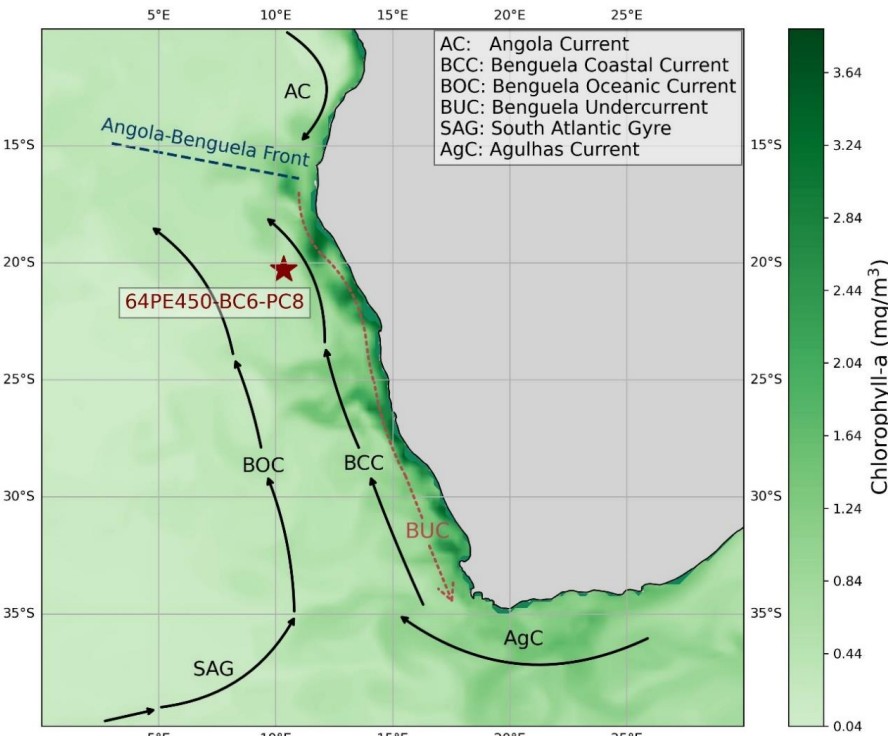

**Figure 2: Map showing the location of sediment core 64PE450-BC6-PC8 and the dominant currents shaping the characteristics of the Benguela Upwelling System. The map is overlain with the distribution of surface water chlorophyll-a concentration of July 2023 obtained from Global Ocean Biogeochemistry**



**Analysis and Forecast (E.U. Copernicus Marine Service Information; https://doi.org/10.48670/moi-00015).**
       **High chlorophyll-a concentrations indicate the high productivity and nutrient-rich upwelled waters of this**
       **region today.**

### 3 Materials and methods

Samples were taken from box core 64PE450-BC6 and piston core 64PE450-PC8 retrieved from the south flank

of the Walvis Ridge, both taken at the same location (approximately -20.29 S, 10.35 E) at a water depth of ~1375

mbss. The box core consisted of 41.5 cm of sediment, whereas the piston core collected 1453 cm (cut into 15

sections, of which we used the first 100 cm within this study). The top of the piston core was missing and, hence,

we combined both the BC and the PC to obtain a near continuous record with a composite depth of 141.5 cm,

which we here refer to as 64PE450-BC6-PC8. The composite record was sampled with a resolution varying

between 2 and 5 cm to optimize coverage of the glacial-interglacial transition. All samples were freeze-dried and

subsequently split in sub-samples to obtain lipid biomarkers and foraminifera from the same core depth.

### 3.1 Foraminiferal sample cleaning

Due to their high abundance in upwelling regions as well as common use in paleoclimate reconstructions (e.g.,

Spero and Lea, 1996), we here selected specimens of *Globigerina bulloides* for the planktonic foraminifera-based

records. Freeze-dried samples were washed over a 63 µm sieve, dried and further dry sieved to separate size

fractions 150-315 µm and 315-425 µm. Specimens of *G. bulloides* were picked from the latter size fraction for

analysis of oxygen and carbon isotopes, minimizing any potential impacts of ontogeny. However, as much more

specimens were needed, the smaller size fraction was used for radiocarbon, element / calcium (El/Ca) and boron

isotope analysis. To construct a benthic foraminiferal carbon isotope record, specimens of *Cibicidoides*

*wuellerstorfi* were picked from the 315-425 µm size fraction.

Foraminiferal samples were cleaned prior to the analysis of El/Ca ratios and stable isotopes, following an adapted

protocol of Barker et al. (2003). This adapted protocol is as follows: for the analysis of the shells' element

concentrations in solution and the boron isotopic composition, specimens were carefully cracked using a scalpel

to open up the chambers and release potential clay content from the inside. The samples were subsequently

transferred to acid cleaned 1.5 mL vials (Treff) and rinsed three times with deionized water (Milli-Q), twice with

methanol, followed by another thorough rinse with deionized water, using ultra-sonification for each rinsing step.

To remove all organic material from the shells, samples were placed in a hot block and oxidized with $NH_4OH$-

buffered 1% $H_2O_2$ solution for 45 minutes at 90 °C. To ensure complete removal or organic material, this step

was repeated up to three times based on visual inspection. After the oxidative cleaning, the samples were

transferred to new pre-cleaned vials (Treff) and leached with diluted acid (1 mM $HNO_3$) followed by rinsing the

samples three times with deionized water. Because the boron isotope analysis is very sensitive to contamination

two additional leaching steps with 1% $NH_4OH$ were added followed by rinsing with deionized water before the

acid leaching. Samples for El/Ca and $\delta^{11}B$ analysis were finally dissolved in 500 µL 0.1 M ultra grade $HNO_3$ and

in 75-80 µL 0.5 M ultra grade $HNO_3$, respectively.



Specimens taken for the analysis of El/Ca ratios with LA-Q-ICP-MS and for the measurement of $\delta^{18}O$ and $\delta^{13}C$ were cleaned following the same clay removal and oxidative cleaning step as described above but without cracking
the shells before the cleaning steps.

### 3.2 Radiocarbon analysis

Radiocarbon analysis ($^{14}C/^{12}C$) on 50-100 specimens of well-preserved shells of *G. bulloides* were performed at Laboratory of Ion Beam Physics, ETH Zürich. The analysis of $^{14}C/^{12}C$ followed the protocol described in Wacker et al. (2013; 2014). Briefly, samples were measured with a gas ion source in a Mini Carbon Dating System
(MICADAS; Synal et al., 2007) with an automated method for acid digestion of carbonates (Wacker et al., 2013). Samples were placed in 4.5 mL exetainers vials (Labco Limited®, UK) and purged with a flow of 60 mL min$^{-1}$ of helium for 10 minutes and subsequently leached with 100 µL 0.02 M ultrapure HCl with an automated syringe to remove adsorbed contaminants. Analysis of the released $CO_2$ from both the leachate and remaining sample provided confirmation for the near complete removal of contaminants. The released $CO_2$ from the leachate was
directly transported by helium to a zeolite trap and injected into the ion source for $^{14}C/^{12}C$ analysis. The remaining leached sample was acidified with 100 µL ultrapure $H_3PO_4$ (85%) and heated at 60 °C for a minimum of 1 hour. The released $CO_2$ was then injected in the ion source for analysis (Wacker et al., 2014; Fahrni et al., 2013). The difference between the radiocarbon values of the leachate and leached samples were less than 5%. Radiocarbon determinations are given in the conventional radiocarbon ages and corrected for isotopic fractionation via $^{13}C/^{12}C$
isotope ratios. Calibration was performed using the Marine20 calibration curve (Heaton et al., 2020) with a local correction to the marine reservoir age ($\Delta R$) of 146 ± 85 $^{14}C$ years (Dewar et al., 2012). These calculations were computed using the Bayesian age-depth model in the Bacon v2.3 package for the R statistical programming software (Blaauw and Christen, 2011).

### 3.3 Analysis of stable oxygen and carbon isotopes

Pre-weighed 20-40 µg of the shells of *G. bulloides* were dissolved in orthophosphoric acid and analysed at 71 °C by a Kiel IV device coupled to a MAT 253 Isotope Ratio Mass Spectrometer (IRMS, Thermo Fischer Scientific®) at the NIOZ. Analyses were calibrated using standard bracketing (NBS-19) and the NIOZ house standard (NFHS-1; Mezger et al., 2016) was used to monitor drift. Accuracy and precision for $\delta^{13}C$ = 1.922 ± 0.05 ‰ and $\delta^{18}O$ = -2.189 ± 0.11 ‰ were calculated across several analytical runs (± 1σ SD, n = 47).

### 3.4 Analysis of foraminiferal Element/Calcium ratios

Prior to the analysis of the samples in solution, a few planktonic foraminifera specimens were screened for preservation to minimize the possibility of diagenetic overprint affecting the geochemical signature of the shells. For this, the ratios of $^{23}Na/^{43}Ca$, $^{24}Mg/^{43}Ca$, $^{25}Mg/^{43}Ca$, $^{27}Al/^{43}Ca$, $^{55}Mn/^{43}Ca$ $^{88}Sr/^{43}Ca$ were simultaneously monitored during the ablation of single chambers of *G. bulloides* by Laser Ablation Quadrupole Inductively
Coupled Plasma Mass Spectrometer (LA-Q-ICP-MS). Laser ablation data was acquired on 60-µm diameter spots with a repetition rate of 4 Hz and a laser energy density of ~1 J cm$^{-2}$. The JCp (*Porites sp.* coral) nano-pellet was used to monitor instrumental drift and JCt (*Tridacna gigas* giant clam; Okai et al., 2004), MACS-3 and the NIOZ



Foraminifera House Standard-2-Nano-Pellet (NFHS-2-NP; Boer et al., 2022) provided further quality control on the measurement. NIST SRM610 was used as calibration standard. Data was evaluated both as profiles and shell

averages.

Approximately 40-50 specimens of *G. bulloides* were dissolved for solution analyses using a Sector-Field Inductively Coupled Plasma Mass Spectrometer (SF-ICP-MS, Thermo Fischer Scientific® Element-2). Applied cleaning procedure is based on Barker et al. (2003) as discussed above. A pre-scan of calcium concentrations

($[Ca^{2+}]$) was performed on an aliquot of 30 µL of the dissolved samples and based on that data subsequently all samples were diluted to match $[Ca^{2+}]$ (100 ppm) for element analyses. Isotopes of $^{11}B$, $^{25}Mg$, $^{138}Ba$ were measured in low resolution and $^{32}S$ in medium resolution to avoid interference. All samples were measured against 4 ratio calibration standards (De Villiers et al., 2002) and alternated with 0.1 M $HNO_3$ in between samples to increase the efficiency of wash-out. All samples are drift-corrected using the NFHS-1 standard (Mezger et al., 2016) and

three additional standards: NFHS-2 (Boer et al., 2022), JCp, and JCt (Okai et al., 2004), to evaluate accuracy and precision of the analytical runs. Uncertainty from the internal precision on the basis of short term stability is < 2 % for B, Mg, Ba, and S. Samples were analysed in replicates yielding an uncertainty of < 3.25 µmol mol$^{-1}$ for B, < 0.02 mmol mol$^{-1}$ for Mg, and S, and < 0.14 µmol mol$^{-1}$ for Ba.

### 3.5 Micro-distillation and boron isotope analysis

Approximately 150 specimens of *G. bulloides* were cleaned for the analysis of boron isotopes. Boron was separated from the calcium carbonate matrix via the micro-distillation technique (Gaillardet et al., 2001; Wang et al., 2010; Misra et al., 2014). 70 µL of the sample was placed on the lid of a Teflon® fin-legged conical beaker (5 mL) and placed upside down on a hotplate at 100 °C for 20-24 hours. The fin-legged vials were wrapped in aluminium foil to provide a heat gradient for a more efficient separation of boron. Once the micro-distillation was

complete, the vials were carefully removed from the hotplate while turning them over and subsequently left for cooling. Sample residue was removed with putting new lids on the beakers and each sample was diluted with 0.2 M HF + 0.2 M $HNO_3$ for a pre-scan of the boron concentration ([B]). Based on the results of the pre-scan, a final dilution was made to set [B] at 5 ppb for the analysis of $\delta^{11}B$.

Analysis of the micro-distilled samples was performed at the NIOZ on a Neptune Plus Multi-Collector Inductively Coupled Mass Spectrometer (MC-ICP-MS, Thermo Fisher Scientific®) equipped with high performance extraction cones (Jet sample cone and 'X' skimmer cone) to maximize sensitivity for boron. Samples were injected using a Savillex® 50 µL min$^{-1}$ C-flow nebulizer and Teflon® Scott type spray chamber. Beams of $^{10}B$ and $^{11}B$ were measured on L3 and H3 Faraday cups equipped with amplifiers using $10^{13}\Omega$ resistors (Misra et al., 2014; Lloyd

et al., 2018). The instrument was tuned to obtain a stable sensitivity, typically 15-25 mV ppb$^{-1}$ B.

Solutions of 0.2 M HF + 0.2 M $HNO_3$ were used for rinsing throughout the analytical run between analyses, and as matrix for each sample and standard. The analysis followed the approach of sample-standard bracketing using NIST 951 as reference standard. All samples and quality control standards were analysed in duplicates and thus

here average values with ± 2σ standard deviations are reported. A coral standard (Chanakya and Misra, 2022) was treated with the complete carbonate cleaning and micro-distillation procedure for each analytical sequence and



repeatedly analysed to monitor long term precision ($\delta^{11}B$ = 24.57 ± 0.65 ‰ 2σ, n = 64). Additionally, non-micro-distilled AE-121 standard was analysed within each run for quality control ($\delta^{11}B$ = 19.48 ± 0.33 ‰ 2σ, n = 46).

In addition to the coral standard, the initial test analysis to validate the boron purification method and instrumental accuracy and precision also included repeated measurements of seawater (Southern Ocean, $\delta^{11}B$ = 39.72 ± 0.25 ‰ 2σ, n = 5) and a boron standard (AE-121) mixed with $CaCO_3$ (trace metal basis, Acros Organics®) to mimic foraminiferal calcium concentrations ($\delta^{11}B$ = 19.53 ± 0.25 ‰ 2σ, n = 18).

**3.6 Estimating past salinity and foraminifera-based temperatures, pH, and $p$CO₂**

Sea surface temperatures (SST) were calculated from foraminiferal Mg/Ca values using the species specific temperature calibration of Mashiotta et al. (1999),

$$Mg/Ca = 0.47\ (\pm0.03)^{0.107(\pm0.003)*SST}, \tag{1}$$

where propagated error was calculated based on 1 standard deviation of the duplicate analysis of Mg/Ca and the uncertainty derived from the calibration equation. Mg/Ca values of planktonic foraminifera are known to be

affected by salinity changes as well (Gray et al., 2018; Dueñas-Bohórquez et al., 2009), however, a correction for this effect requires an independent estimate for salinity and a species-specific calibration. The same is true for the effect of pH (Gray et al., 2018; Gray and Evans, 2019). Moreover, as both effects are relatively minor and adding them would also introduce additional uncertainties, we here decided to refrain from correcting for salinity and/or pH when calculating past temperatures.


For calculating past carbonate chemistry, salinity is an important parameter and it was estimated based on its conservative relationship with relative sea level change (Waelbroeck et al., 2002). Modern seawater salinity of the BUS (35.43 ± 0.30) was derived from the WOCE Global Data Version 3.0 (Schlitzer, 2000) based on the five closest datapoints to the location of core 64PE450-BC6-PC8.


The measured $\delta^{11}B$ values of *G. bulloides* were converted into pH (Hemming and Hanson, 1992) using Eq. (2):

$$pH = pK_B - \log(-(\delta^{11}B_{sw} - \delta^{11}B_{borate}) / (\delta^{11}B_{sw} - \alpha * \delta^{11}B_{borate} - \varepsilon), \tag{2}$$

where the equilibrium constant, $pK_B$ (Dickson, 1990), was calculated for each sample based on SST derived from the Mg/Ca values of *G. bulloides* and salinity based on sea level. The fractionation factor between $B(OH)_3$ and

$B(OH)_4^-$, expressed here as α, is 1.0272 ± 0.0006, from which fractionation, ε, is derived as 27.2 ± 0.6 (Klochko et al., 2006). Boron isotopic composition of seawater, $\delta^{11}B_{sw}$ is 39.61 ± 0.2 ‰, based on a large range of temperature, salinity and depth conditions (Foster et al., 2010), and the $\delta^{11}B$ of borate was calculated from the measured $\delta^{11}B$ of *G. bulloides* using the species specific core-top calibration from Raitzsch et al. (2018).

Uncertainty on the reconstructed pH value was determined for each sample through error propagation that considered the above described uncertainties of $pK_B$, α, ε, $\delta^{11}B_{sw}$ and the standard deviation (external uncertainty) based on duplicate or triplicate analysis of foraminiferal $\delta^{11}B$.



Concentrations of $CO_2$ are based on pH and inorganic carbon chemistry calculations using the PyCO2SYS
package (Humphreys et al., 2022) in Python version 3.11.2. Uncertainty is propagated for each computed carbon
chemistry parameter as described in Humphreys et al. (2022).

**3.7 Lipid extraction and alkenone analyses**

Lipids were extracted from the freeze-dried and homogenized sediment samples using an accelerated solvent
extractor (ASE® 350, DIONEX®) at the NIOZ. Samples were extracted with dichloromethane (DCM) and
methanol (9:1, $v/v$) at 100 °C to obtain the total lipid content and subsequently dried under $N_2$ gas at 35 °C in a
Caliper TurboVap LV Evaporator. Samples were then redissolved in DCM and run through an $Na_2SO_4$ column to
eliminate excess water. The extract was passed through an alumina ($Al_2O_3$) column and separated into apolar,
ketone and polar fractions using a mixture of hexane : DCM (9:1, $v/v$), hexane : DCM (1:1, $v/v$), and DCM :
methanol (1:1, $v/v$), respectively. All extracts were dried under $N_2$ and the ketone fraction was further utilized to
obtain the relative abundance and $\delta^{13}C$ values of the long chain alkenones.

Ketone fractions were dissolved in ethyl acetate and concentrations of alkenones were measured using a gas
chromatograph with flame ionization detection (GC-FID, Agilent® 6890N) equipped with silica capillary column
(CP-Sil 5 CB; 50 m x 0.32 mm, 0.12 μm film thickness). The temperature program of the GC-FID analyses used
an initial temperature of 70 °C that increased with a rate of 20 °C min⁻¹ to 200 °C followed instantly by heating at
a rate of 3 °C min⁻¹ until it reached 320 °C where it remained constant for 10 minutes.

Based on the initial concentration measurement, samples were diluted with ethyl acetate to allow stable carbon
isotope analysis using a gas-chromatography-isotope ratio-mass spectrometer (GC-IRMS, Thermo Fisher
Scientific® Delta V Advantage Trace® 1310). The GC-IRMS was equipped with crossbond trifluoropropylmethyl
polysiloxane columns (Rtx-200; 60m x 0.32, 0.5 μm film thickness) and helium as a carrier gas. Each sample was
manually injected on the GC-IRMS. The starting temperature of the GC-IRMS was 70 °C which then increased
with 18 °C min⁻¹ until reaching 250 °C. After reaching that temperature heating continued with 1.5 °C min⁻¹ until
320 °C, where it was kept stable for 25 minutes. Samples were analyzed for carbon isotopes in duplicates and
instrumental accuracy was monitored through measurement of the of B5 $n$-alkane mixture standard (provided by
A. Schimmelmann, Indiana University) every day (i.e. after every 6-7 samples). The isolink II combustion reactor
was oxidized for 10 minutes every day before the start of standard and sample analysis. Each analysis was
followed by 2 minutes of seed oxidation to maintain the reactor oxygenated.

**3.8 Calculation of alkenone based temperatures and $p$CO₂**

Alkenone-based sea surface temperatures were derived from ketone unsaturation index ($U^{K'}_{37}$), where $U^{K'}_{37}$ is
defined as the relative abundance of di- and tri-unsaturated $C_{37}$ methyl alkenones (Prahl and Wakeham, 1987):

$$U^{K'}_{37} = C_{37:2} / (C_{37:3} + C_{37:2}). \tag{3}$$

Sea surface temperature was then calculated using the alkenone temperature calibration model developed for the
Atlantic region (Conte et al., 2006).



Fractionation of stable carbon isotopes during photosynthesis ($\varepsilon_{p37:2}$) can be computed based on the difference between the carbon isotopic ratio of aqueous carbon dioxide ($\delta^{13}C_{CO2}$) and the organic biomass ($\delta^{13}C_{org}$):

$$\varepsilon_{p37:2} = [(\delta^{13}C_{CO2} + 1000) / (\delta^{13}C_{org} + 1000) - 1] \times 1000. \tag{4}$$

$\delta^{13}C_{CO2}$ was derived from the carbon isotopes of planktonic foraminifera, *G. bulloides* ($\delta^{13}C_p$) corrected for the temperature dependent fractionation during calcite precipitation (Romanek et al., 1992) and the fractionation between dissolved and gaseous carbon dioxide (Mook et al., 1974).

$\delta^{13}C_{org}$ was calculated from the carbon isotopes of di-unsaturated alkenones ($\delta^{13}C_{37:2}$) as

$$\delta^{13}C_{org} = [(\delta^{13}C_{37:2} + 1000) / (1 - \Delta\delta^{13}C_{org})] - 1000, \tag{5}$$

where $\Delta\delta^{13}C_{org}$ expresses the carbon isotopic difference between $C_{37:2}$ and DIC, that has been defined between 3 – 6 ‰ based on culture experiment (Riebesell et al., 2000; Schouten et al., 1998; Van Dongen et al., 2002). We here take the commonly applied value of 4.2 ‰ (Bijl et al., 2010; Pagani et al., 2005; Pagani et al., 2010; Pagani et al., 2011; Seki et al., 2010; Palmer et al., 2010).

Based on $\varepsilon_{p37:2}$, aqueous $CO_2$ ($[CO_2]_{aq}$) can be reconstructed as followed (Hayes, 1993; Pagani et al., 2002):

$$[CO_2]_{aq} = b / (\varepsilon_f - \varepsilon_{p37:2}), \tag{6}$$

where $\varepsilon_f$ stands for the carbon isotopic fractionation associated with carbon fixation estimated as 25 ‰ (e.g., Popp et al., 1998). Parameter $b$ expresses all physiological factors affecting total carbon isotope fractionation that includes cell shape and size, membrane permeability as well as the algae's growth rate (Jasper et al., 1994; Rau et al., 1996; Popp et al., 1998; Conte et al., 1994; Riebesell et al., 2000). Earlier studies using phytane (Bice et al., 2006; Damsté et al., 2008) and alkenone (Witkowski et al., 2018) to reconstruct $pCO_2$ estimated $b$ for a mean value of 165 - 170 ‰ kg µM⁻¹. As growth rate and thereby nutrient availability have a large influence on the physiological factors and, accordingly, $b$ values are highly correlated to $[PO_4^{3-}]$ (Bidigare et al., 1997), $b$ can be best described at our core site by estimating past changes in $[PO_4^{3-}]$ (Pagani et al., 2005). Here, $[PO_4^{3-}]$ is estimated based on the barium over calcium ratio (Ba/Ca) of planktonic foraminifera, *G. bulloides* (Lea and Boyle, 1989; Lea and Boyle, 1990b, a; Martin and Lea, 1998; Lea and Boyle, 1991). We therefore constrain past changes in b as:

$$b = [118.52 \times (Ba/Ca \times [PO_4^{3-}]_{modern} / Ba/Ca_{modern}) + 81.42]. \tag{7}$$

Average modern $PO_4^{3-}$ concentration ($[PO_4^{3-}]_{modern}$) in the BUS is 0.63 µmol kg⁻¹ (obtained from GLODAPv2023; Lauvset et al., 2024; Olsen et al., 2016; Key et al., 2015) whereas the corresponding modern foraminiferal Ba/Ca value ($Ba/Ca_{modern}$) was analyzed here (19.08 µmol mol⁻¹). Eq. (7) basically assumes a constant and proportional relation between Ba and $[PO_4^{3-}]$. This seems reasonable for our purposes as surface water Ba concentration has been shown to be reflected proportionally in foraminiferal Ba/Ca (Lea and Boyle, 1991; Hönisch et al., 2011) and the cold nutrient rich surface waters are generally enriched in dissolved barium (e.g., Davis et al., 2020).

To calculate atmospheric $pCO_2$ from aqueous concentrations of $CO_2$, Henry's law was applied using the temperature and salinity dependent solubility constant, $K_0$.



$$pCO_2 = [CO_2]_{aq} / K_0. \hspace{4cm} (8)$$

Uncertainty propagation for the calculated $pCO_2$ values was based on the errors derived from 1 standard deviation of duplicate analysis of $\delta^{13}C$ of the extracted ketone fraction, $\delta^{13}C$ of foraminifera and Ba/Ca values of foraminifera. The largest uncertainty in alkenone based $pCO_2$ reconstructions originates from the estimated past $[PO_4^{3-}]$ and to incorporate potential variation in nutrient levels during the deglaciation, an additional uncertainty of 0.2 µmol kg$^{-1}$ was assigned to the known modern values of $[PO_4^{3-}]$. This uncertainty is based on the gradient measured today in upper 50 meters of the water column, which is more than the variability observed in surface water today, but also includes potential changes in the upwelled waters.

## 4 Results

### 4.1 Radiocarbon ages

The calibrated mean radiocarbon ages generally increase with depth in both BC and PC cores used here. Sediment core 64PE450-BC6 comprises 41.5 cm, where the core-top sample was dated at 4.863 ± 0.284 ka BP, and an age of 9.551 ± 0.248 ka BP at 40 cm bsf (Fig. 3 a). This suggests an average sedimentation rate of about 0.01 cm yr$^{-1}$, with somewhat higher values (> 0.01 cm yr$^{-1}$) at the top 12 cm. Alternatively, the upper 10 cm bsf have constant ages due to bioturbation. Radiocarbon analyses from sediment core 64PE450-PC8 included 6 samples of the upper 100 cm of sediment collected. The age-depth model for this core suggests 9.994 ± 2.042 ka BP years at 5 cm bsf depth (Fig. 3 b), which is likely due to the loss of sediment at the top (common during piston coring). Low sedimentation rates (0.002 cm yr$^{-1}$) characterize the top 10 cm bsf of this core which is approximately in line with the sedimentation rate at the deepest parts of box core 64PE450-BC6 (~0.006 cm yr$^{-1}$). However, average sedimentation rate in 64PE450-PC8 is lower (0.004 cm yr$^{-1}$) compared to the average values observed in the box core, which, in part, might also be due to compaction with increasing depth. The top 60 cm bsf of the core shows a steady increase in sedimentation rate (0.002 – 0.007 cm yr$^{-1}$) and therefore the low average values may be attributed to a relatively abrupt decrease in sedimentation rate at 60 cm bsf in the core, which correspond to an age of 23.586 ± 0.410 ka BP. Between 60 and 100 cm bsf depth sedimentation rates remain 0.001 – 0.002 cm yr$^{-1}$. Due to the uncertainty related to sediment deposition at these depths and the low sedimentation rate, further analysis of this study focuses only on the 41.5 cm bsf of the core 64PE450-BC6 and upper 65.5 cm bsf of core 64PE450-PC8, which together comprise a near-continuous time interval from ~5 to 27 ka BP .





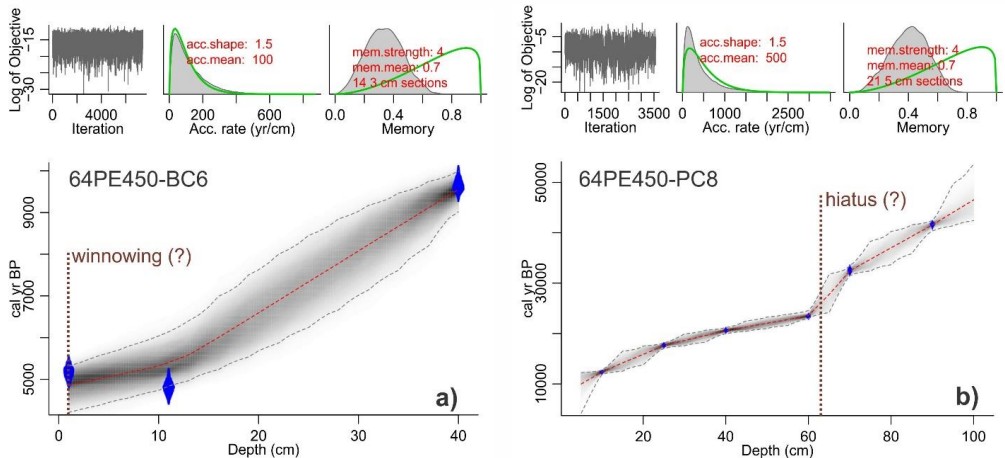

**Figure 3: Age-depth model of a) 64PE450-BC6 (box core) and b) 64PE450-PC8 (piston core) based on radiocarbon dates, where blue diamonds indicate the sampling depth for $^{14}$C analysis. The calibration of radiocarbon ages and the figure was generated using the Bacon v2.3 package for the R statistical programming software (Blaauw and Christen, 2011). Calibration was performed using the Marine20 calibration curve (Heaton et al., 2020) with a local carbon reservoir correction (ΔR) of 146 ± 85 $^{14}$C years (Dewar et al., 2012). Red dashed lines show mean values of the best fitted model and grey dashed lines indicate 95% confidence interval. Note, that both panel a) and b) show the complete sediment records acquired by the box corer and piston corer, respectively, but due to potential hiatus in the piston core as indicated in panel b), only the top 65.5 cm of 64PE450-PC8 was utilized for temperature and carbon system reconstruction in this study.**

**4.2 Stable isotopes**

Carbon isotope values of the planktonic foraminifer, *G. bulloides,* vary between -1.4 ‰ and 0.6 ‰ (VPDB). The glacial part of the record is marked by relatively high δ$^{13}$C values with a maximum of 0.6 ‰ at 23.3 ka BP. After that, there is a rapid decrease to a minimum value of -1.1 ‰ at 22 ka BP, then it stabilizes around -0.8 ‰ until 18 ka BP (Fig. 4 a). The δ$^{13}$C values of the benthic foraminifer, *C. wuellerstorfi*, although measured at somewhat lower resolution, range between 0.5 ‰ and 1.0 ‰. It appears that there is a continuous increase in benthic δ$^{13}$C from 27 ka BP until the most recent sample (Fig. 4 a).

The δ$^{18}$O values of *G. bulloides* range from 0.0 ‰ (VPDB) to 3.2 ‰ with the most depleted values at ~8 ka BP (Fig. 4 b). During the glacial, δ$^{18}$O values decreased from the highest values at 23.6 ka BP (peak glacial), showing rapid changes during the deglaciation, subsequently decreasing again gradually before reaching a plateau at about 10 ka BP. During the Holocene these values stayed relatively stable and varied only between 0.0 ‰ and 0.6 ‰ (VPDB). The trend differs from the lower resolution benthic record of δ$^{18}$O measured on *C. wuellerstorfi* (Fig. 4 b). The benthic foraminiferal δ$^{18}$O values are consistently higher compared to the *G. bulloides* values, which is in line with lower bottom water temperatures. This difference, however, appears smaller during the end glacial than during the Holocene.



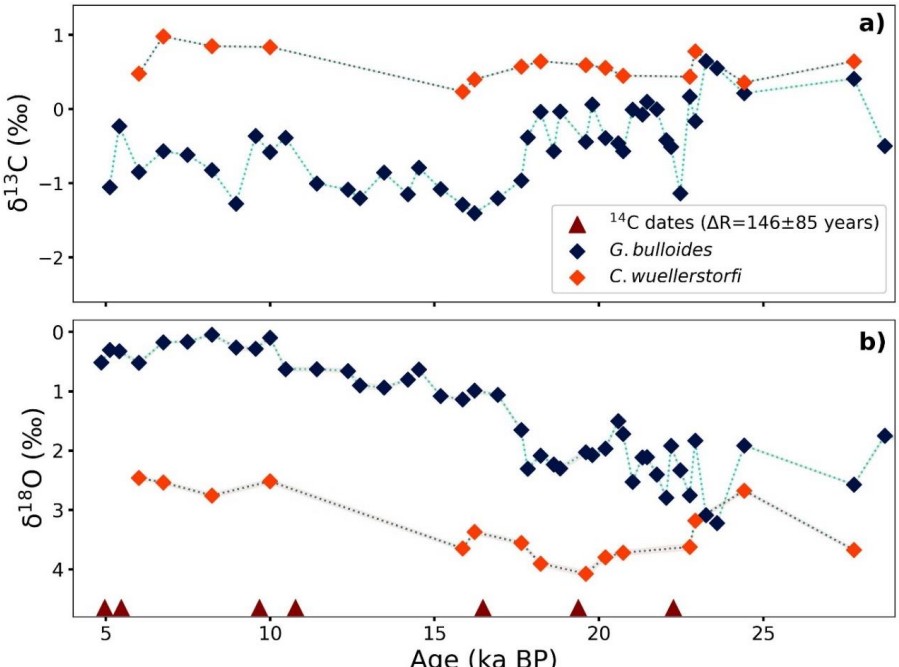

**Figure 4: a) δ$^{13}$C and b) δ$^{18}$O values of planktonic (*G. bulloides*) and benthic (*C. wuellerstorfi*) foraminifera**
**plotted with the age model. Red triangles indicate the ages tied with radiocarbon dates.**

The boron isotopic composition of the planktonic foraminifer, *G. bulloides,* ranges between 15.1 and 17.0 ‰
(relative to NIST 951) with larger variations during the last 6-5 ka BP (Fig. 5 a). The lowest δ$^{11}$B values were
observed at 5.4 ka BP, whereas δ$^{11}$B values reach a maximum at 13.5 ka BP. Prior to this maximum value, δ$^{11}$B
values show an increasing trend from 27.8 to 13.5 ka BP (Fig. 5 a).

The carbon isotopic composition of the alkenones shows its heaviest value (-22.4 ‰) at 19.6 ka BP. After this
peak, δ$^{13}$C values reach a minimum (-23.4 ‰) at 15.9 ka, then increasing again towards the most recent values (-
22.7 to -22.9 ‰; Fig. 5 b).

**4.3 Element/Ca ratios in the planktonic foraminifer, *G. bulloides***

The general trend in foraminiferal B/Ca values shows an increase from glacial to recent (Fig. 5 c). The highest
B/Ca value (52 μmol mol$^{-1}$) is observed in the top of the record and the lowest value (35 μmol mol$^{-1}$) at 22.8 ka
BP. Somewhat higher variability is observed between 22.8 and 15.2 ka BP, with B/Ca values ranging between 35
and 48 μmol mol$^{-1}$.




S/Ca values show a trend opposite to that observed for B/Ca. While S/Ca shows a maximum (1.33 mmol mol$^{-1}$) at 18.6 ka BP, it decreases to 0.93 mmol mol$^{-1}$ at 5.1 ka (Fig. 5 d). Overall the record shows a gradual change in foraminiferal S/Ca, without much scatter.

The oldest part of the record shows relatively stable Ba/Ca values at around 6 µmol mol$^{-1}$. During the last 15 ka BP, however, more variability is observed for Ba/Ca (Fig. 5 e). The here observed trends are not resembling the trends observed for either B/Ca or S/Ca. However, the overall increasing trend from 27 ka BP to present day Ba/Ca values somewhat coincides with the trend in B/Ca.

Mg/Ca reaches maximum values of 2.85 and 2.81 mmol mol$^{-1}$ at 16.9 and 6.7 ka BP, respectively (Fig. 5 f). Substantially lower values characterise the interval between 16.9 and 6.7 ka BP, when Mg/Ca ranges between 2.46 and 2.69 mmol mol$^{-1}$. The lowest values (2.30-2.40 mmol mol$^{-1}$) were found at 4.9-5.4 ka BP and 21.3-27.8 ka BP.

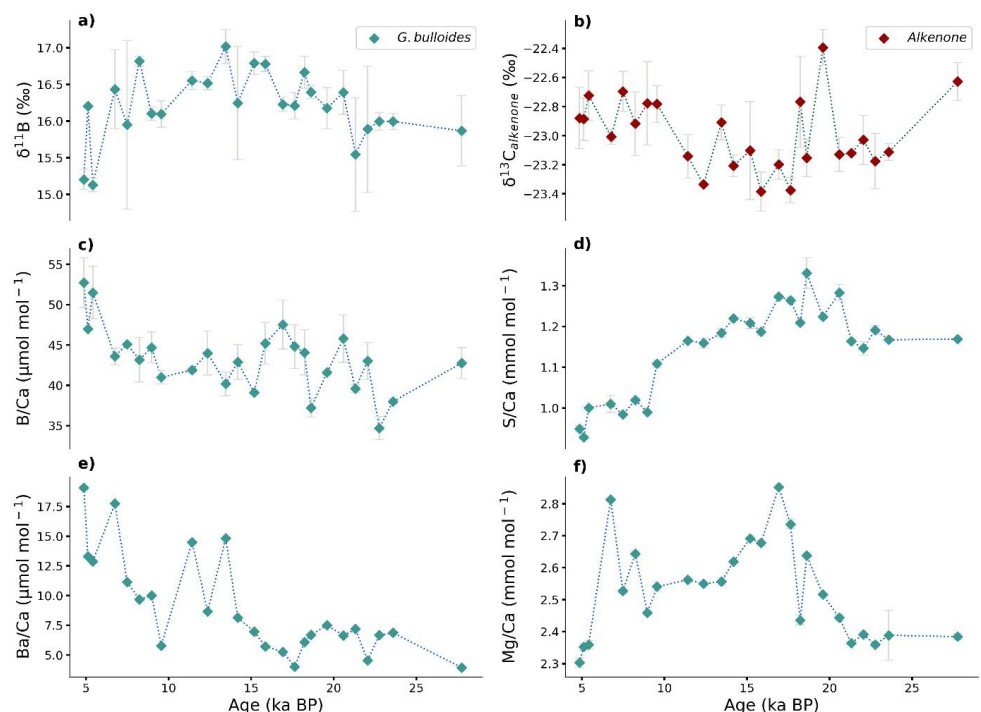


**Figure 5: Measured a) foraminiferal δ$^{11}$B, b) alkenone δ$^{13}$C, and foraminifera element concentration: c) B/Ca, d) S/Ca, e) Ba/Ca, f) Mg/Ca plotted over the past 27 ka BP at the Benguela Upwelling System. Error bars show ± 1σ standard deviation. When error bar is not show, the error of the duplicate measurement is smaller than the symbol.**


**4.4 U$^{K'}_{37}$ sea surface temperatures**



The alkenone based $U^{K'}_{37}$ record shows continues increase throughout the deglaciation (Supplementary Fig. S1). The lowest value (0.56) was measured at 27 ka BP which increases to 0.58 until 26.3 ka BP. The interval of 22.0 - 19.6 ka BP records again low $U^{K'}_{37}$ values, which then follows a steady increase until 9.6 ka BP. During the early

Holocene $U^{K'}_{37}$ values stabilize around 0.7 until 6.8 ka BP when values slightly start to decrease reaching 0.67 in the uppermost part of the record.

## 5 Discussion

### 5.1 Local temperatures over the LGM, last deglaciation, and Holocene

The proxy-based temperature reconstructions indicate most of the well-known Southern Hemisphere climate

events, such as found within the $\delta^{18}O$ ice core record from EPICA-Dome C (EDC; Jouzel et al., 2007) for the last 27 ka BP (Fig. 6).

The $U^{K'}_{37}$ based sea surface temperature reconstruction shows low temperatures (18.2 – 18.5 °C) between 23.6 and 18.6 ka BP (Fig. 6 a) corresponding to the last glacial maximum (LGM; e.g., Clark et al., 2009; Hughes et al.,

2013). Oxygen isotopes from the shells of *G. bulloides* show more enriched values during the same period and hence indicate low sea surface temperatures, in agreement with $U^{K'}_{37}$, however, with significantly more scatter in $\delta^{18}O$ values (1.5 – 3.2 ‰). Foraminiferal $\delta^{18}O$ values not only depend on temperature, but also on the stable oxygen isotopic signature of the water and hence salinity. Meltwater is transported to large distances with the AAIW, that will result in larger regional impact in the upwelling zone than in the global ocean. Therefore, shifts

in e.g. sea ice extension or river runoff, might also have impacted local seawater $\delta^{18}O$ and hence foraminiferal oxygen isotopes. Similarly, multiple studies suggested an overall reduced Agulhas leakage during the LGM compared to deglacial levels (Pether, 1994; Flores et al., 1999; Rau et al., 2002; Peeters et al., 2004; Charles and Morley, 1988; Wefer et al., 1996; Franzese et al., 2006), and changes therein could also have resulted in enhanced $\delta^{18}O$ variability during the LGM.


Our record shows a rapid warming following the onset of deglaciation from ~17.8 ka BP observed in both the alkenone based SST reconstruction and $\delta^{18}O$ values of *G. bulloides* (Fig. 6 d). The last glacial termination (approximately 18 ka BP) resulted in the extensive release of meltwater on the Northern Hemisphere, weakening Atlantic Meridional Overturning Circulation (AMOC; Mcmanus et al., 2004; Rahmstorf, 2002; Denton et al.,

2010; Hodell et al., 2017; Pöppelmeier et al., 2023). This, in turn changed global heat distribution and hence the cooling in the North Atlantic region known as the Heinrich Stadial I (e.g., Bond et al., 1993; Cacho et al., 1999; Mcmanus et al., 1994), which was accompanied by a warming in the Southern Hemisphere (e.g., Calvo et al., 2007; Wang et al., 2013).

Following the Heinrich Stadial I, further, more gradual temperature increase during the deglaciation is evident from the $U^{K'}_{37}$ temperatures, while only a slight decrease in $\delta^{18}O$ values is observed between 14.6 and 13.0 ka BP. The Antarctic Cold Reversal (ACR) during the Bølling-Allerød Northern Hemisphere warming (e.g., Pedro et al., 2016; Blunier et al., 1997), resulted in either no change (Lamy et al., 2007) or a slight decrease in SST (Vandergoes



et al., 2008). Moreover, here we observed only minor changes between the trends in alkenone (increasing trend)
and the foraminiferal Mg/Ca based SST (decreasing trend; Fig. 6 b) and the $\delta^{18}$O (decreasing trend) signal. Offsets
likely reflect differences in either seasonality between coccolithophores and *G. bulloides* (Leduc et al., 2010), or
differences in water depth where the signals were recorded. Hence minor differences between the records could
be due to a shift in productive season and or a shift in depth habitat. Because offsets are only minor (< 4.4 °C)
unravelling these signals is highly speculative.

At approximately 12.7 ka BP, we see a slight increase in both $U^{K'}_{37}$ and $\delta^{18}$O based temperatures, which is in line
with the general Southern Hemisphere temperature record. The Younger Dryas (Alley, 2000 and references
therein) started around this time with another meltwater discharge into the North Atlantic, weakening overturning
circulation, and leading to simultaneous warming in the Southern Hemisphere.

The most recent part of the record, the early Holocene, shows a cooling trend following the 8.2 ka BP event known
from the Northern Hemisphere (Matero et al., 2017; Barber et al., 1999; Thomas and Via, 2007; Morrill et al.,
2013). Although represented by only one datapoint in our record, the observed warming at around 8 ka BP (here
and in Ljung et al., 2008) could be assigned to the well-known cooling in the Northern Hemisphere in line with
the AMOC slow down during this event (Barber et al., 1999).

The SST reconstruction based on foraminiferal Mg/Ca values is generally in line with the $U^{K'}_{37}$ and $\delta^{18}$O record,
confirming overall trends. Absolute values, however, differ, which is likely due to depth differences between the
proxy signal carriers. Alkenone-based temperature reconstruction agrees with reported SSTs from the upper 50 m
of the Northern Benguela (Santana-Casiano et al., 2009), while the foraminifer based temperature (Mg/Ca)
corresponds to the values observed somewhat deeper (100-150 m, GLODAPv2023; Lauvset et al., 2024;
Supplementary Fig. S2). Vertical dispersion of *G. bulloides* may be large, but this suggested living habitat
corresponds well with previously reported living depths for this species (Tapia et al., 2022; Lessa et al., 2020;
Rebotim et al., 2017). Whereas the LGM and the deglaciation are clearly visible in Mg/Ca, $U^{K'}_{37}$, and the $\delta^{18}$O
records, during the Holocene the Mg/Ca temperature record clearly deviates from the other two. The offset
between $U^{K'}_{37}$ and foraminiferal Mg/Ca values during the LGM is about 3.0 °C, and this gradually increases to 5.6
°C during the early Holocene. This implies that foraminiferal Mg/Ca values are affected by other factors than SST
alone. Partial dissolution of foraminiferal shells at depth may also affect Mg/Ca values and thereby reconstructed
temperatures through the preferential loss of $Mg^{2+}$ (Dekens et al., 2002; Regenberg et al., 2006). However, impact
of dissolution is probably minor only as the water depth at site 64PE450-PC8-BC6 is less than 1.4 km and *G.
bulloides* is reported to be less sensitive to dissolution compared to other surface dwellers (Mekik et al., 2007).
Whereas Mg/Ca values may also be affected by early diagenesis (Hover et al., 2001; Kozdon et al., 2013; Ni et
al., 2020; Panieri et al., 2017; Sexton et al., 2006; Stainbank et al., 2020) El/Ca ratios in the shell's profile obtained
through laser ablation did not show any evidence for such diagenetic effects (Karancz et al., 2024b).

Nevertheless, the trends observed in the different proxies seem to accurately represent the changes known for the
Southern Hemisphere, albeit that foraminiferal Mg/Ca based temperatures may be affected by some changes in
depth habitat or seasonality of *G. bulloides*.

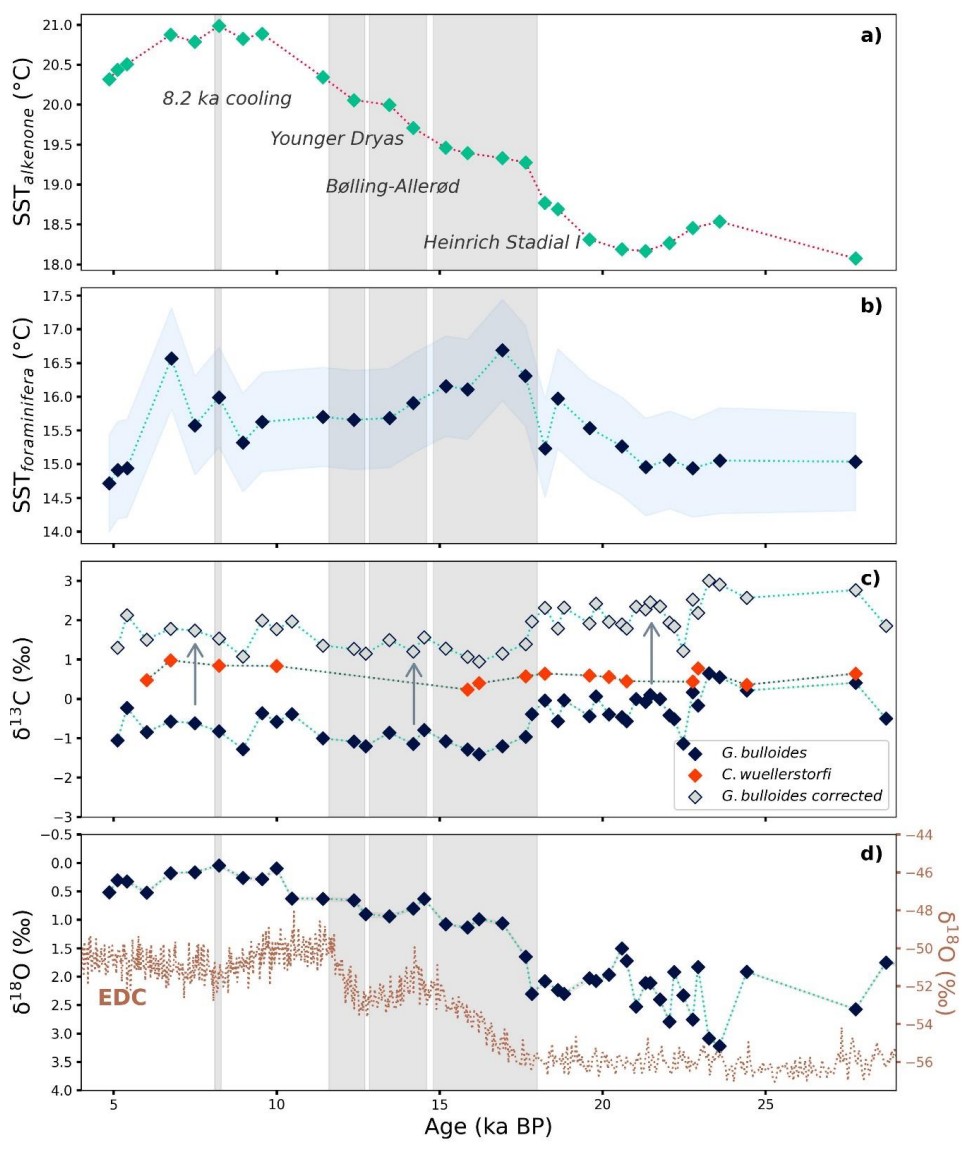


**Figure 6: Reconstructed sea surface temperatures (SST) based on a) the alkenone unsaturation index, U$^{K'}_{37}$, and b) foraminiferal Mg/Ca, c) δ$^{13}$C analysed in benthic (*C. wuellerstorfi*) and planktonic (*G. bulloides*) foraminifera with corrected values, and d) δ$^{18}$O of planktonic foraminifera and δ$^{18}$O ice core record from EPICA-Dome C (EDC; Jouzel et al., 2007) shown for the past 27 ka BP. Modern day SST at core site 64PE450-BC6-PC8 is approximately 20.7 °C (GLODAPv2023; Lauvset et al., 2024; Santana-Casiano et al., 2009). Grey shaded areas mark climate events as labelled on the uppermost panel. Blue shaded area in panel b) indicates the error propagated from temperature calibration uncertainty and ± 1σ standard deviation of the duplicate measurement of the samples. Analysis of the stable isotopes (panel c and d) provided an error smaller than the symbols shown on the figure. Arrows in panel c) indicate the direction of the correction on the planktonic foraminiferal δ$^{13}$C.**





**5.2 Biological carbon pump**

Comparing benthic (*C. wuellerstorfi*) and planktonic (*G. bulloides*) trends in $\delta^{13}$C shows they have similar values during the last glacial and higher $\delta^{13}$C values for the benthic than for the planktonic foraminifera during the
interglacial. This is in contrast to what one would expect if BCP determined the foraminiferal carbon isotope signatures (Hain et al., 2014; Hilting et al., 2008). Generally, DIC in surface waters is enriched in $^{13}$C as the BCP results in preferential export of $^{12}$C rich organic matter to the deeper water masses, where it is released through remineralization. The efficiency and strength of the BCP is known to be affected by multiple processes, such as the formation, sinking, and interaction of aggregates with other minerals (Fowler and Knauer, 1986; Alldredge
and Silver, 1988; Armstrong et al., 2001; Francois et al., 2002; Klaas and Archer, 2002; De La Rocha and Passow, 2007; Turner, 2015), and the efficiency is generally reflected by the offset in $^{13}$C between the surface and deep water ($\Delta\delta^{13}$C). However, species specific offsets from equilibrium values between seawater DIC $\delta^{13}$C and foraminiferal carbonate $\delta^{13}$C can challenge interpretation of the $\Delta\delta^{13}$C and are likely responsible for the here observed lower $\delta^{13}$C values of *G. bulloides* compared to *C. wuellerstorfi*.


Application of foraminiferal $\Delta\delta^{13}$C as a proxy for BCP efficiency requires a direct relation with the $\delta^{13}$C of DIC. Benthic foraminiferal $\delta^{13}$C values, and in particular $\delta^{13}$C values of the epifaunal *C. wuellerstorfi*, are generally considered faithful recorders of the $\delta^{13}$C values of DIC, with carbonate $\delta^{13}$C values being close to equilibrium (Thomas and Shackleton, 1996; Hilting et al., 2008). The stable carbon isotopic composition of DIC today is
approximately 0.5 – 0.7 ‰ at a depth of 1.3 km along the latitude 20° S (Kroopnick, 1980; Kroopnick, 1985; Sarnthein et al., 1994; Curry and Oppo, 2005; Schmittner et al., 2013) which agrees well with the value inferred from *C. wuellerstorfi* in the uppermost sample (6.0 ka BP) of core 64PE450-BC6-PC8.

Stable carbon isotopic values from planktonic foraminifera have been shown to be generally lower with respect
to the equilibrium values of DIC (Kahn, 1979; Kahn and Williams, 1981; Oppo and Fairbanks, 1989; Spero, 1992), indicating a strong biological impact (i.e. the vital effect; e.g., Spero, 1992; De Nooijer et al., 2014; Erez, 2003). For instance, symbiont-bearing species such as *O. universa* and *T. sacculifer* show offsets in $\delta^{13}$C as much as 1.8 and 1.4 ‰ respectively, depending on irradiance level (Spero, 1992; Spero and Lea, 1993). Although *G. bulloides* lacks algal symbionts (Hemleben et al., 1989) it has been shown to deviate even more from the ambient
seawater $\delta^{13}$C values (Kahn and Williams, 1981; Spero and Lea, 1996). Several factors likely add together to the observed offset, such as carbon chemistry ([$CO_3^{2-}$]; 0.3-1.0 ‰; Spero et al., 1997; Bijma et al., 1999), temperature, (2.4-2.6 ‰; Bemis et al., 2000) and respiration (Zeebe et al., 1999). More recently, Bird et al. (2017) suggested that also bacterial symbiosis may partly explain the observed offset for $\delta^{13}$C in *G. bulloides*. While symbiont photosynthesis contributes to elevating foraminiferal $\delta^{13}$C due to preferential $^{12}$C removal, geochemical signature
of *G. bulloides* is more likely to be controlled by the respiration of photoautotrophic cyanobacteria that produces depleted $CO_2$ and hence, decreases shell $\delta^{13}$C values (Bird et al., 2017). Irrespective of the process involved, a substantial correction has to be applied to the $\delta^{13}$C values of *G. bulloides* to approach the heavier seawater DIC $\delta^{13}$C values. Increasing temperature (Bemis et al., 2000) and [$CO_3^{2-}$] (Bijma et al., 1999) will also increase the offset between $\delta^{13}$C values of *G. bulloides* and seawater [DIC] suggesting that larger corrections are required



during the Holocene than during the last glacial. Still, when the corrections for changes in temperature and $[CO_3^{2-}]$ are applied, trends remain the same, showing the highest $\delta^{13}C$ values of planktonic foraminifera during the LGM (Supplementary Fig. S3). Hence, despite the uncertainties in interpreting the absolute planktonic $\delta^{13}C$ values, the trend in $\Delta\delta^{13}C$ should still provide a measure for changes in the efficiency of the BCP. We here applied a constant equilibrium offset of 2.4 ‰ (Fig. 6 c) which is the average correction required based on temperature variation in
our record (Bemis et al., 2000). This value is close to the offset of the $\delta^{13}C$ value of the core-top sample with the modern $\delta^{13}C$ values of the DIC (Kroopnick, 1985).

Offsets between the $\delta^{13}C$ of the planktonic and benthic foraminifera reflect differences in the BCP, but potentially also changes in the dominant water mass at the cores' locations. Intermediate depths of the South Atlantic are
dominated today by the Antarctic Intermediate Water and this likely remained the major water mass over the last glacial cycle (Pahnke et al., 2008; Howe et al., 2016; Gu et al., 2017). However, it is unclear whether the depth range of the AAIW increased (Muratli et al., 2010) or decreased (Ronge et al., 2015; Li et al., 2021) during the LGM compared to present day. In the western Atlantic, $\delta^{13}C$ values of benthic foraminifera suggest persistence of AAIW water masses at the depth of our core site (e.g., Curry and Oppo, 2005). As our values correspond to
those found by Curry and Oppo (2005) we suggest also here a sustained influence of southern water masses, with the $\delta^{13}C$ value of DIC in the AAIW during the LGM remaining relatively similar to present day. The analysed $\delta^{13}C$ values of benthic foraminifera from the LGM in this study show on average values that are 0.2 ‰ lower values than those during the Holocene.

Variations in the stable carbon isotopic composition of surface seawater DIC are attributed to the changes in biological activity and air-sea exchange (Lynch-Stieglitz et al., 1995). While enhanced biological activity will result in an increase of $\delta^{13}C$ values of DIC, more intense air-sea exchange will contribute to a decrease. In upwelling regions, the upwelled light carbon may still result in a net decrease in $\delta^{13}C$ values despite the enhanced biological activity.


Summarizing the observed impacts, assuming a more or less stable presence of AAIW at intermediate depth and after correcting for offsets, a larger difference between planktonic and benthic foraminiferal $\delta^{13}C$ values during the LGM compared to the Holocene is evident (Fig. 6 c; and Supplementary Fig. S3), suggesting a more efficient BCP.

**5.3 $p$CO$_2$ record of the BUS over the last 27 ka BP**

Although $\delta^{11}B$ in foraminifera shells and $\delta^{13}C$ of alkenones are the most commonly applied methods to reconstruct $p$CO$_2$, these proxies are only very rarely compared in the same record. Since these proxies are recording different components of the speciation of carbon in seawater and have different biases, they not necessarily have to show similar results. Here, we observe only a modest change in pH (8.08-8.23, derived from foraminiferal $\delta^{11}B$) during
the last deglaciation, whereas $p$CO$_2$ values show a change from 180 to 280 ppm (derived from $\delta^{13}C$ of alkenones) (Fig. 7).



Minor variability in pH was reported previously by Raitzsch et al. (2018) for the Walvis Ridge for the same time interval, although reconstructed pH values were slightly higher (0.10-0.14 pH units). Although only minor, the offset is in line with the core studied here being closer to the upwelling area, as the major upwelling area extends only about 200 km out of the coast today (Lutjeharms and Meeuwis, 1987; Lutjeharms and Stockton, 1987). With lowest pH values in the core of the upwelling area and values increasing towards the open ocean, the trend in the offset between the two areas is minor but in the right direction.

Using pH and total alkalinity, $p$CO$_2$ can be calculated, suggesting higher $p$CO$_2$ compared to the known atmospheric values over the past 27 ka BP (Fig. 7; Petit et al., 1999). Calculated $p$CO$_2$ based on the foraminiferal $\delta^{11}$B only match seawater equilibrium values at 13.5 and 8.2 ka BP, corresponding to two major climate events (the Bølling-Allerød event and the 8.2 ka BP cooling event, respectively). Whereas the Bølling-Allerød event corresponds to maximum AMOC, the 8.2 ka BP event is generally assumed to be associated to a reduced AMOC due to the meltwater input in the North Atlantic (Matero et al., 2017; Barber et al., 1999; Pedro et al., 2016; Blunier et al., 1997). However, the calculated $p$CO$_2$ values are associated with considerable uncertainty for a large part related to total alkalinity being ill-constrained. Using estimates of total alkalinity based on relative sea level change is debated (e.g., De La Vega et al., 2023) as this does not account for all changes in alkalinity on glacial-interglacial timescales. Because total alkalinity calculated based on the relative sea level change during the last deglaciation results in only a small (less than 10 ppm) offset, we here used constant alkalinity (2349 ± 11 µmol kg$^{-1}$, GLODAPv2023; Lauvset et al., 2024) in combination with boron isotope-based pH to determine $p$CO$_2$. The trends observed here are not affected by the alkalinity values used.

The seawater $p$CO$_2$ reconstruction based on the $\delta^{13}$C of alkenones follows past atmospheric $p$CO$_2$ known from the Vostok ice core record remarkably well (Petit et al., 1999). This suggests that over the interval studied here, this remained more or less in equilibrium with the atmosphere with regard to CO$_2$ and did not act as an appreciable source or sink. An offset is observed (about 65 ppm) during the Holocene, between 11 and 7 ka BP, when alkenone-based reconstruction suggests somewhat lower $p$CO$_2$ values compared to the ice core record, although this difference falls well within the uncertainty of the proxy. This could indicate a temporary transition of the area to a CO$_2$ sink as the seawater becomes undersaturated with respect to CO$_2$.

Previous studies have observed discrepancy between alkenone based $p$CO$_2$ reconstruction and ice core records (Palmer et al., 2010; Andersen et al., 1999; Zhang et al., 2013; Witkowski et al., 2020; Jasper et al., 1994), which could be related to disequilibrium between sea surface and the atmosphere, especially at dynamic sites like upwelling regions. However, it may also be explained by the mechanism of CO$_2$ uptake in the algal cell. Current application of the alkenone $p$CO$_2$-proxy assumes the passive diffusion of CO$_2$ (Bidigare et al., 1997; Laws et al., 1995). However, much evidence indicates that alkenone-producers adapted a carbon concentrating mechanism (CCM; Stoll et al., 2019; Reinfelder, 2011; Bolton and Stoll, 2013; Badger, 2021), which enables carbon acquisition in the cell through the active pumping of HCO$_3^-$ to the chloroplast during low $p$CO$_2$ conditions. While such mechanism inevitably hampers the application of the $p$CO$_2$-proxy, efficiency of the CCM is potentially also affected by local conditions which may result in alkenone based $p$CO$_2$ reconstructions at some sites still being



able to reproduce glacial atmospheric values (Jasper and Hayes, 1990; Bae et al., 2015). Furthermore, it has been shown that CCM's may induce their own isotopic fractionation (e.g., Wilkes and Pearson, 2019).

Also, $p$CO$_2$ reconstructions based on alkenones δ$^{13}$C values are subject to uncertainties related to the $b$ factor. The $b$ value expresses the effect of multiple parameters related to the physiology of the alkenone producers (Jasper et al., 1994; Rau et al., 1996; Popp et al., 1998), which is best represented by a linear relationship to nutrient availability (Bidigare et al., 1997). Often modern, constant, [PO$_4^{2-}$] is assumed to estimate the $b$ factor for reconstructing $p$CO$_2$ (Pagani et al., 1999; Zhang et al., 2013; Pagani et al., 2005; Witkowski et al., 2020).

Assuming that the membrane permeability has not changed significantly, one can correct for growth rate effects of the alkenone producers (Zhang et al., 2019; Zhang et al., 2020). We here relied on the analysis of Ba/Ca in planktonic foraminiferal shells as a proxy for seawater [PO$_4^{3-}$]. Ba/Ca is suggested to reflect nutrient variations (Lea and Boyle, 1989; Lea and Boyle, 1990b, a; Martin and Lea, 1998) but does not vary with temperature, salinity or carbon chemistry parameters (Lea and Spero, 1994; Hönisch et al., 2011) unlike other suggested nutrient

proxies such as Cd/Ca (Oppo and Rosenthal, 1994; Allen et al., 2016) and Zn/Ca (Van Dijk et al., 2017b). Thus, using this approach represents an efficient way to tackle uncertainties originating form local conditions. While uncertainties described for the above mentioned factors may impact minor changes in the $p$CO$_2$ record, observed trends on glacial-interglacial timescales can be still interpreted.

Uncertainties in the different factors affecting $p$CO$_2$ estimates urge to develop additional independent carbon system parameter proxies that can supplement the δ$^{11}$B-pH proxy. Among the six variables, the most commonly tackled parameter is [CO$_3^{2-}$] using element / calcium ratios (El/Ca) such as B/Ca (Allen and Hönisch, 2012; Allen et al., 2012; Haynes et al., 2017; Yu and Elderfield, 2007; Yu et al., 2007) and more recently suggested S/Ca (Van Dijk et al., 2017a) and S/Mg (Karancz et al., 2024a). Alternative foraminifera based $p$CO$_2$ reconstructions using

core top calibrations for B/Ca (Krupinski et al., 2017) and S/Mg (Karancz et al., 2024a) are discussed in the Supplementary Material (Supplementary Text S1). Before unravelling differences in local carbon uptake and/or outgassing over glacial-interglacial cycles, it is important to decide which foraminifer-based proxy records best reflects local carbon speciation. Although using B/Ca (Supplementary Fig. S4) and S/Mg (Supplementary Fig. S5) bring alkenone and boron-based $p$CO$_2$ reconstructions closer together, inherent error propagation also renders

the records more difficult to differentiate. Hence, although we note the necessity and possibilities to improve B-isotope based $p$CO$_2$ reconstructions, we here base our down-core foraminifera-based $p$CO$_2$ reconstruction on using boron isotopes only.



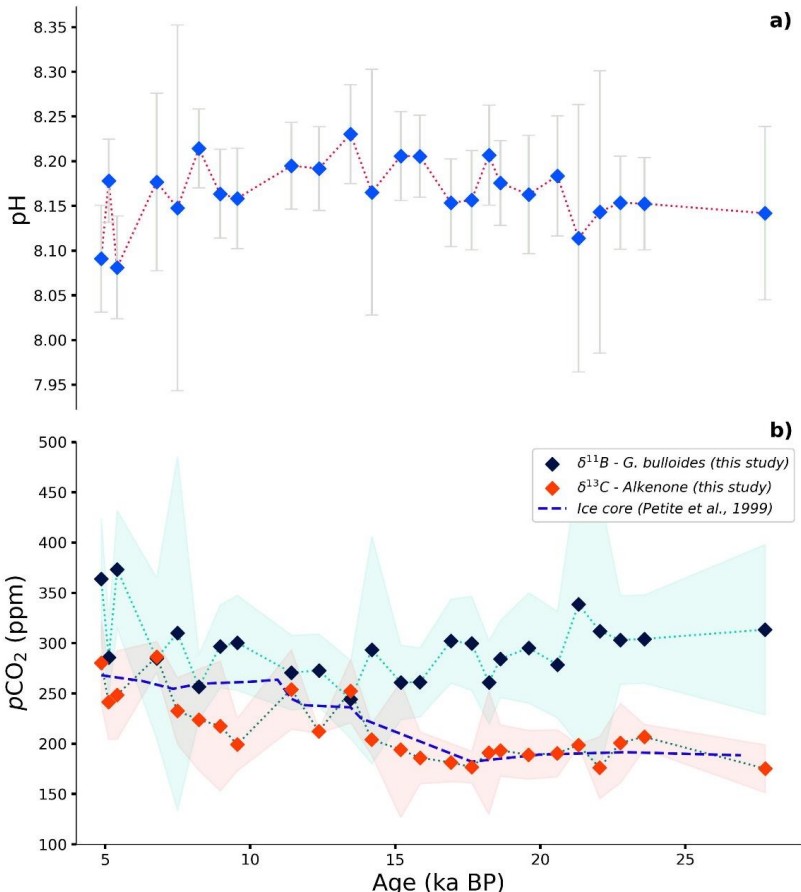

**Figure 7: Reconstruction of a) pH based on δ$^{11}$B of *G. bulloides* and b) pCO₂ based on δ$^{11}$B of *G. bulloides* combined with a constant total alkalinity value of 2349 ± 11.07 μmol kg$^{-1}$ (dark blues diamonds) and δ$^{13}$C of alkenones (red diamonds). Modern day *p*CO₂ value of the AAIW is approximately 326 ppm (Lauvset et al., 2024; Salt et al., 2015). Blues dashed line shows the Vostok ice core record of *p*CO₂ (Petit et al., 1999). Light green and red shaded area represent propagated error for the foraminifera and alkenone based reconstructions, respectively. See further details on uncertainty propagation in the text.**

**5.4 Change in the efficiency of BCP and CO₂ disequilibrium**

Most obvious from comparing the alkenone and foraminifera based *p*CO₂ reconstruction is the difference in amplitude of change on a glacial-interglacial time scale. Whereas the alkenone based reconstruction closely mimics atmospheric changes, the foraminifera-based reconstruction shows a constant *p*CO₂. Because the *G. bulloides* are proliferating during the upwelling season, they likely primarily reflect the somewhat deeper upwelled water compared to alkenones which are formed by the surface dwelling coccolithophorids. This implies more intense drawdown and recycling of CO₂ from the surface layers to the intermediate waters. The water



upwelled in the BUS is AAIW and hence a relative increase in $p$CO$_2$ in the upwelled waters implies enhanced
CO$_2$ storage in these waters.

Lower atmospheric CO$_2$ during the glacial is likely explained by multiple processes. The larger extent of sea ice over the glacial Southern Ocean prevented CO$_2$ escaping from seawater in an area today acting as a major CO$_2$ exchange region (Stephens and Keeling, 2000), whereas enhanced iron fertilization likely contributed to more
efficient utilization and transport of carbon and nutrients to the deep (Martin, 1990; Martínez-García et al., 2014). Aeolian transport and dissolution in the shelf regions might have provided important sources of iron at that time (Martin, 1990; Tian et al., 2023). Also locally, aeolian transport presumably increased due to the intensified trade winds (Stuut et al., 2002), although a more humid climate may (Stuut et al., 2002; Cockcroft et al., 1987) or may not (Shi et al., 1998; Partridge et al., 1999) have prevailed in southwest Africa during the LGM. Although stronger
winds may have provided sufficient iron for phytoplankton growth locally, excess iron input in the sub-Antarctic region most likely provided a much larger source for additional mid-depth CO$_2$ storage (Martínez-García et al., 2014). A more efficient biological carbon pump, as indicated by the offset between the planktonic and benthic foraminiferal carbon isotope records, suggest that an increased supply of carbon in the upwelling areas from intermediate depths to the surface, may have been effectively counterbalanced.


Concentrations of CO$_2$ in the intermediate waters show on average comparable values during the Holocene and LGM, and hence imply a difference in the $p$CO$_2$ gradients ($\Delta p$CO$_2$ = $p$CO$_{2(\text{intermediate})}$ - $p$CO$_{2(\text{surface})}$) between the surface and intermediate waters (Fig. 8). Interglacial difference between alkenone and foraminifera-based reconstruction shows $\Delta p$CO$_2$ value of about 62 ± 29 ppm, while during glacial times this difference increases to
approximately 104 ± 29 ppm. Atmospheric $p$CO$_2$ was significantly reduced during the LGM, hence the presence of an increased amount of CO$_2$ at intermediate depths implies either enhanced upwelling or that the upwelled waters were richer in CO$_2$ or both. Foraminifera-based proxies indicate more intense upwelling during glacial times (Oberhänsli, 1991; Little et al., 1997), but at the same time radiolarian based upwelling proxies suggest reduced upwelling (Des Combes and Abelmann, 2007). Due to its location and the influence of water masses both
from the north and the south, cells of the BUS are characterized by different environmental conditions (e.g., temperature and nutrients). During the LGM, cold source waters likely impacted the northern cells of the BUS more than its central and southern parts (Des Combes and Abelmann, 2007) affirming complexity of this upwelling system. While we may conclude that upwelling intensities were different from one cell to another, potentially also impacted by the offshore transition of the modern strong upwelling cells (e.g., Mollenhauer et al.,
2002), increased cold water input does not necessarily correlate with stronger upwelling (Des Combes and Abelmann, 2007), potentially explaining conflicting interpretations.



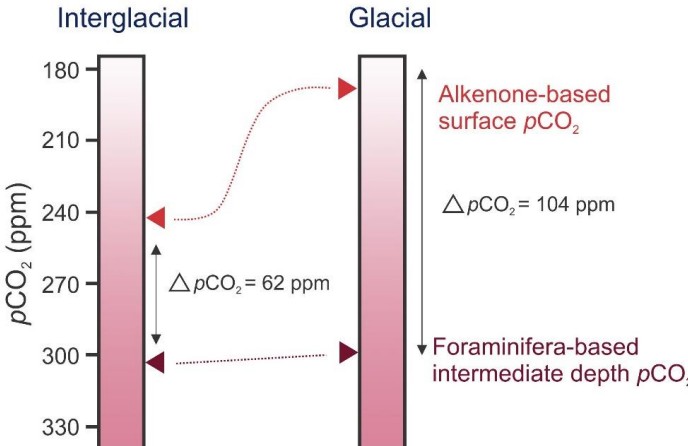

**Figure 8: Schematic comparison of interglacial and glacial $pCO_2$ values. Red arrows mark average interglacial and glacial values calculated from the alkenone and planktonic foraminifera (pH and total alkalinity) based proxies of this study.**

Based on comparing $pCO_2$ proxies, with *G. bulloides* recording primarily the upwelled waters and alkenones the surface waters, we see evidence for enhanced storage of carbon at depth during the glacial. This is in line with the reconstructed high productivity on the basis of elevated iron input in the high nutrient, low chlorophyll (HNLC) areas at that time in the sub-Antarctic region (Martínez-García et al., 2014). The resulting mid depth high $CO_2$ waters provide also at that time the source for upwelled waters in the BUS, which could have resulted in the local release of (part of the) stored $CO_2$ if not prevented by an efficient biological carbon pump. If iron input was enhanced also locally (Stuut et al., 2002), this could increase the biological pump, which then acted as an effective cap on the stored carbon. Both remote and local enhanced iron supplies hence contributed to lowering atmospheric $pCO_2$ during the glacial.

**6 Conclusions**

Carbon system proxies were applied to demonstrate changes in inorganic carbon chemistry of the Northern Benguela Upwelling System over the last 27 ka BP. Temperature reconstructions based on both organic and inorganic proxies indicate that the BUS is generally associated to climatic changes observed in the Southern Hemisphere. While surface values of $pCO_2$ reconstructed from $\delta^{13}C$ of alkenones follow atmospheric changes of $pCO_2$ remarkably well, the foraminifera-based reconstructions suggest minor variation in $pCO_2$ at intermediate depth since the last glacial maximum until present. Hence, the increased gradient of $pCO_2$ between the surface waters and depth observed for the last glacial period provide evidence for enhanced storage of carbon in the Antarctic Intermediate Waters. Outgassing of $CO_2$, however, could be effectively prevented by the biological carbon pump as also indicated by the offset in the $\delta^{13}C$ of planktonic and benthic foraminifera.



**Data availability**

All data used in this study can be obtained from the NIOZ Data Archive System at https://doi.org/10.25850/nioz/7b.b.lh (Karancz et al., 2024b): Table 1. Uncalibrated radiocarbon ages; Table 2. Single foraminifera analysis (LA-Q-ICP-MS); Table 3. $\delta^{18}O$ and $\delta^{13}C$ of planktonic and benthic foraminifera, El/Ca and $\delta^{11}B$ of planktonic foraminifera, $U^{K'}_{37}$ and $\delta^{13}C$ of alkenones.

**Author contributions**

SK, LJdN, SS, GJR designed the study. RH and ZE collected the sample material and SK prepared and processed the samples. SK, BwdW, MTJvdM, SM were responsible for the analysis of foraminifera and alkenones. JL and NH conducted the radiocarbon analysis. SK interpreted and visualized the data under the supervision of LJdN, SS, GJR and drafted the paper with contribution from all co-authors.

**Competing interests**

The authors declare that they have no conflict of interest.

**Acknowledgement**

This work was carried out under the program of the Netherlands Earth System Science Centre (NESSC), financially supported by the Ministry of Education, Culture and Science (OCW) and the European Union's Horizon 2020 research and innovation program under the Marie Skłodowska-Curie grant with agreement No. [847504]. We thank Wim Boer, Piet van Gaever, Patrick Laan, Jort Ossebaar, Anchelique Mets, and the laboratory of Ion-Beam Physics at ETH Zurich for the technical support during sample preparations and the geochemical analysis.

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
