# Peer review of "Contrasts in the marine inorganic carbon chemistry of the Benguela Upwelling System since the Last Glacial Maximum"

_EGUsphere, 2024_

## Author Comment (AC1)

Review of Karancz et al., "Glacial-interglacial contrasts in the marine inorganic carbon chemistry of the Benguela Upwelling System," submitted to *Climate of the Past,* by Jesse Farmer

Karancz and colleagues present a reconstruction of ocean carbonate chemistry and its proximal drivers in the Benguela Upwelling System (BUS) covering the last ~27 kyr based on a combination box-piston core from the Walvis Ridge. I commend the authors for applying a wide range of analytically challenging proxies in their attempt to understand past carbonate chemistry in the BUS. The manuscript is overall well written, especially the detailed and extensive methods, and the top-line interpretation/conclusion of increased intermediate ocean carbon storage based on the difference between alkenone $\varepsilon_P$ and boron-based $pCO_2$ proxies is quite novel.

For the bad news: I found several deficiencies that temper my enthusiasm for publication. Given that I think some major rewrites (and possibly additional data) will be needed, I recommend major revisions that allow for sufficient time to address this. I want to emphasize that, given the quality and quantity of data here already, I fully expect a revised paper would be worthy of publication in *Climate of the Past*.

**Response:** Hereby we would like to thank you for the valuable comments which helped to clarify and improve the manuscript. Below we provide answers to all comments point by point and indicate how they will be implemented.

**Major comments.**

**1. Core stratigraphy.** I found the combined stratigraphy from BC6 and PC8 to be lacking. If I read this correctly, the authors took the length of BC6 (41.5 cm) and appended PC8 to the bottom to create a composite depth scale of 141.5 cm. While this is probably in the general ballpark given the radiocarbon dates, this approach is far too simplified. The authors would have to be incredibly lucky to have it such that the total recovery of the box core (41.5 cm) just happened to align with the total amount of lost material from the top of the piston core! (In my experience, this has never once happened).

To address this, the authors should use physical properties measurements on the two cores to align them in depth space and create a composite depth scale. This can be any measurable property – bulk density, reflectance/color, XRF, etc. Once they have this data, they can then align the two cores based on these properties and create a single depth-age model in rBacon using the $^{14}C$ dates vs. composite depth.

**Response:** Thank you for pointing this issue out and we agree that the creation of the composite record on a time scale requires stronger underpinning and rephrasing for clarity. In the adjusted manuscript we will refrain from using a composite depth scale and, instead, discuss the individual age models of the BC and PC records, and try to align those appropriately. The total dated length of BC6 needs to be corrected in the manuscript text to be 40.59 cm (figures and interpretations are not affected, as the age model is based on the 40.59 cm length), providing a continues record of 4.863 to 9.636 ka BP. From sediment core PC8, only the top 65.5 cm was considered in this study based on the radiocarbon dates and the rBacon age-depth model, as there is potentially a hiatus below this depth (as illustrated in Figure 3). The top 4.24 cm of PC8 is disturbed likely due to the piston coring.

We used, as suggested by the reviewer, brightness (l*) from the core scanner to align the two cores. This way we show that there is some overlap between BC6 and PC8, based on detailed line-scan pictures of the two cores. As the reviewer already indicates, the cores do obviously not exactly follow, but have an overlap of 4.24 cm interval (~5.360 to 9.635 ka BP). This implies that, the top 4.24 cm of PC8, which is disturbed likely due to the coring technique, overlaps with 30.14 cm of BC6 (i.e., from 10.45 to 40.59 cm), suggesting severe compression of the top of PC8 (and absence of material above that). Deformation at the top of piston

cores is not unusual, and much more likely than the box core being disturbed. Hence, this overlap can be used to align and tie together the two age models of BC6 and PC8 to produce a near-continues record of 27.75 ka BP. Although the records are clearly very well correlated, we decided to use the box core at the top section as the compressed section (even with the very high correlation observed) might be somewhat disturbed. The manuscript text will be modified accordingly, including this new information. The figure showing the alignment of the two cores based on reflectance data will be included as a Supplementary Figure:

[Figure]

*Figure S1. Light reflectance (l\*) measured in sediment cores 64PE450-BC6 (black line) and 64PE450-PC8 (red and green lines) plotted over the last 27 ka BP. Panel a) shows the record including the overlap of 64PE450-BC6 with the top 4.24 cm of 64PE450-PC8, which has been excluded from this study due to sediment disturbance likely caused by the coring technique used (green line). Panel b) shows the alignment of 64PE450-BC6 and 64PE450-PC8 that produces a near continues record from 27 to 4.8 ka BP.*

Moreover, for the reviewer, we here also present the alignment based on Ca/Ti ratio measured by XRF-scanning of both cores. Although the measured ratios of PC8 align well with those measured in BC6, the age model cannot be confirmed by this data alone due to the lower resolution of the XRF-scanning analysis (1-cm resolution) compared to the color line-scan data (63-μm resolution). Hence, we will add only the line-scan data to the corrected manuscript.

[Figure]

*Additional Figure for the reviewer. Log(Ca/Ti) ratio (XRF-scanning element intensities) of 64PE450-BC6 (black line) and 64PE450-PC8 (red line and green circles) plotted over the last 27 ka BP. The top 4.24 cm of 64PE450-PC8 marked with green circles is not included in this study.*

I don't ask for new data lightly in what is already an impressive multiproxy work. But there is very little other than $^{14}$C (and then, only six of the eight shown dates) to benchmark this chronology. What else exists does not inspire much confidence – the benthic $\delta^{18}$O record (Figure 4b) is effectively missing the entire deglacial section and so is of minimal utility. And I do not see at all the assignment of millennial-scale deglacial events in Figure 6 (it should be noted that *G. bulloides* $\delta^{18}$O does not look like EDC). To this end, significant age model refinement will be needed if the authors wish to discuss millennial-scale features in their records during the deglaciation.

**Response:** We agree that correlations on a sub-Milankovitch time scale cannot be made with confidence here. For the overall aim of the paper, these detailed correlations are not needed as we mainly compare glacial to interglacial differences. Still, as we have the data at quite a high resolution, in the original manuscript we compared our data to existing records.

We noticed the resemblance of our records to the Southern Hemisphere climate responses based on the gradual increase of $U^{K}_{37}$ temperatures from 23 ka BP onwards. This increase is also visible in a parallel decrease in $\delta^{18}$O values for *G. bulloides*. The early deglaciation is to our opinion very much like that observed on the Southern Hemisphere. Comparing our record with both northern and southern hemisphere records, we also notice that individual climate events show similarities to the trends observed in Northern Hemisphere records, suggesting that the location of our sediment core was affected by both Northern and Southern Hemisphere processes. Acknowledging that our records miss the features necessary to make detailed correlations for individual events, we still would like to allow the readers to appreciate potential correlations for themselves. Hence in the revised manuscript, we will also add (to Figure 6) a key Northern Hemisphere record (NGRIP) and adjust the text accordingly within our discussion. We do believe that our independent age model acquired through $^{14}$C-dating allows to at least compare large-scale deglacial trends and potentially multi-millennial changes within our record to key climate records from both hemispheres.

[Figure]

*Adjusted Figure 6: Reconstructed sea surface temperatures (SST) based on a) the alkenone unsaturation index, $U^{K'}_{37}$, and b) foraminiferal Mg/Ca, c) $\delta^{13}C$ analysed in benthic (C. wuellerstorfi) and planktonic (G. bulloides) foraminifera with corrected values, d) $\delta^{18}O$ of benthic (C. wuellerstorfi) and planktonic (G. bulloides) foraminifera, and e) $\delta^{18}O$ ice core record from EPICA-Dome C (EDC; Jouzel et al., 2007) and North Greenland Ice Core Project (NGRIP; North Greenland Ice Core Project members, 2004) shown for the past 27 ka BP. Modern day SST at core site 64PE450-BC6-PC8 is approximately 20.7 °C (GLODAPv2023; Lauvset et al., 2024; Santana-Casiano et al., 540 2009). Grey shaded areas mark climate events as labelled on the uppermost panel. Blue shaded area in panel b) indicates the error propagated from temperature calibration uncertainty and ± 1σ standard deviation of the duplicate measurement of the samples. Analysis of the stable isotopes (panel c and d) provided an error smaller than the symbols shown on the figure. Arrows in panel c) indicate the direction of the correction on the planktonic foraminiferal $\delta^{13}C$ values.*

**2. Presentation of alkenone ε_P-derived pCO_2.** Keeping in mind my affinity toward boron isotope approaches, the presentation of alkenone ε_P-derived $pCO_2$ stuck me as rather outdated. Namely, the approach outlined in the methods Section 3.8 does not acknowledge any of the vigorous discussion around the feasibility of this approach or its potential limitations. (Note that the authors do start to tackle this much further down on L651-678). I think the authors need to be more up front about what is known regarding the limitations of this proxy and how they sought to address these. Do haptophyte CCMs matter in an upwelling region? Is Ba/Ca actually a functional phosphate proxy in an upwelling region, where high OM remineralization rates can lead to $BaSO_4$ precipitation? Even if that problem is overcome, is Ba/Ca-derived $PO_4$ by itself enough of a constraint on the physiology of the alkenone producing community to address these issues?

Ultimately, it does matter that the authors' ε_P-derived $pCO_2$ is in reasonable agreement with ice core $pCO_2$, and the authors could use this to argue that these recognized proxy complications may not be as significant in this particular setting. But the complications must be addressed head-on, and the reasoning for the authors' choices on calculation of $pCO_2$ should be made explicit to the reader.

**Response:** We acknowledge issues and uncertainties with using the $\delta^{13}C$ of alkenones in a low-[$CO_2$] time interval, and apply this proxy here as an additional, independent method to reconstruct $pCO_2$. Accordingly, we will add a section to introduce limitations of this approach within the Introduction section.

Sensitivity of the alkenone-based $pCO_2$ reconstructions to the 'b' factor (i.e., all physiological parameters affecting carbon isotope fractionation) has been tackled in various ways to improve the application of this proxy. While the approaches include compensation for e.g., nutrient levels, cell size and growth rate, these parameters do not necessarily correlate with each other (e.g., Riebesell et al., 1993), which makes it complicated to estimate carbon isotope fractionation during photosynthesis.

We agree with the reviewer that even though these uncertainties are known, they may not apply to this particular setting considering the good agreement between the alkenone-based $pCO_2$ and the ice core $pCO_2$ reconstructions. Results of this study using the Ba/Ca ratio as a nutrient proxy suggests, that the application potential of these various approaches may be regionally different, and the primary mechanisms controlling the 'b' factor need to be reevaluated for distinct environments, such as upwelling regions. Adaptation of active carbon uptake through CCM in haptophyte algae has been shown to potentially hamper the use of ε_p in $pCO_2$ reconstructions under low, for instance glacial, [$CO_2$] conditions (Badger et al., 2021). However, the mechanisms and controls on how coccolithophores apply CCM is not fully constrained (e.g., Reinfelder, 2011) and CCM activity may also vary between species (e.g., Goudet et al., 2020).

The reviewer is referring to Ba remobilization as a complicating factor for using Ba. This applies to sedimentary Ba. Whereas here we use Ba measured on calcite shells. The correlation between seawater Ba and shell Ba has been published by e.g., Hönisch et al. (2011) and is not affected as long as the shells are adequately cleaned.

**3. B isotope results, S/Mg and calculation of pCO_2.** There are a few issues here:

o  *G. bulloides size fractions and their $\delta^{11}B$.* Numerous studies indicate that different size fractions of planktic foraminifera possess different B isotope ratios (Hönisch and Hemming, 2004; Henehan et al., 2013, 2016). Although I do not think this has been demonstrated explicitly in *G. bulloides*, the community tends to work in quite limited size fractions when measuring B isotopes in *G. bulloides*: 300-355 µm (Martinez-Boti et al., 2015) or 315-355 µm (Raitzsch et al., 2018, cited in the manuscript). In

this study, however, size fractions are not constrained; it appears the authors used specimens from 150 to 425 μm (Section 3.1). This adds an additional source of uncertainty that should be propagated into uncertainty on the reconstructed pH (L285), and it may be quite significant (in excess of 1‰, Henehan et al. 2013 Figure 6).

**Response:** We are aware of potential impacts of ontogenetic variability on $\delta^{11}B$. Yet, unpublished data of Paulhac Buisson et al. (under review) suggest that $\delta^{11}B$ in *G. bulloides* is not affected by the foraminifer's size and confirms that the ontogenetic variability of $\delta^{11}B$, as presented from various species before, derives from the symbiont's activity. The impact of ontogenetic variability on foraminiferal $\delta^{11}B$ values has been shown for *T. sacculifer* (Hönisch and Hemming, 2004), *G. ruber* (Henehan et al., 2013), and *O. universa* (Henehan et al., 2016) which are all symbiont-bearing planktonic foraminifera. In their microenvironment, the pH is affected by the symbionts' physiological processes, such as respiration and photosynthesis, and the degree of pH alteration depends on symbiont abundance and density, and hence size of the foraminifera. We here used *G. bulloides*, which is a symbiont-barren species. We will add a statement about this to the Methods section.

o *Poor precision in some G. bulloides $\delta^{11}B$ replicates*. Looking at Figure 5a, there are a few datapoints where the replicate precision on the sample is > ±0.5‰. Boron is a tricky isotope system to measure, so I wonder if there was some plasma instability during these measurements. But regardless, the authors may wish to exclude these data as they don't appear sufficiently well-constrained to be useful for calculating pH or pCO2.

**Response:** While seasonal differences of *G. bulloides* may contribute somewhat to a higher uncertainty in the pH reconstruction, we agree that datapoints with replicate precision as high as ±0.5‰ gives only an estimate and do not provide sufficient constraints on the $p\mathrm{CO_2}$ reconstruction. These datapoints will be removed accordingly: this removal, however, does not affect the interpretation of the results.

[Figure]

*Adjusted Figure 7: Reconstruction of a) pH based on $\delta^{11}B$ of G. bulloides and b) pCO$_2$ based on $\delta^{11}B$ of G. bulloides combined with a constant total alkalinity of 2349 ± 11.07 µmol kg$^{-1}$ (dark blue diamonds) and $\delta^{13}C$ of alkenones (red diamonds). Modern day pCO$_2$ of the AAIW is approximately 326 ppm (Lauvset et al., 2024; Salt et al., 2015). Blue dashed line shows the Vostok ice core record of pCO$_2$ (Petit et al., 1999). Light green and red shaded area represent propagated error for the foraminifera and alkenone based reconstructions, respectively. See further details on uncertainty propagation in the text.*

o  *Using carbonate ion alongside pH to constrain the carbonate system.* If I follow the authors' approach correctly, they seek to employ *G. bulloides* S/Mg or B/Ca as a separate constraint on the carbonate system to address uncertainty in paleoalkalinity (L680 and supplement). Unfortunately, this does not work well for a reason not included in the study: carbonate ion and pH strongly covary in the modern ocean ranges of alkalinity and DIC (see Figure 10 in Rae et al., 2011). For this reason, error propagation alone is not sufficient; the true uncertainty in the carbon system parameters derived using pH from B isotopes and carbonate ion from S/Mg, B/Ca would need to account for covariance of these parameters. Given this, I think it would be better just to remove this text/approach.

**Response:** Thank you for pointing this out. We agree and, therefore, the application of S/Mg and B/Ca for pCO$_2$ reconstruction will be removed from this study.

**4. $\delta^{13}C$ gradients and estimation of BCP.** In section 5.2, the authors do a pretty good job of laying out the complications to using $\delta^{13}C$ gradients as a proxy for the biological carbon pump (see also section 4.4 in Farmer et al., 2021). After laying out these significant complications, they state on L611-615 "a larger difference between planktonic and benthic foraminiferal $\delta^{13}C$ values during the LGM compared to the Holocene is evident (Fig. 6 c; and Supplementary Fig. S3), suggesting a more efficient BCP". After re-reading this section, I do not agree with this conclusion. The data could more simply be explained as a change to lower $\delta^{13}C$ in AAIW source waters due to inefficient air-sea CO$_2$ exchange in a seasonally sea-ice covered Southern Ocean. To remove this influence, the authors could difference their benthic $\delta^{13}C$ record from one of similar depth in the South Atlantic not under the influence of Benguela Upwelling.

**Response:** We agree that variation in the water mass source present at the core location is an important aspect when foraminiferal $\delta^{13}C$ values are evaluated. Not only seasonal air-sea exchange but also organic matter decay and consequently lower pore water $\delta^{13}C$ may impact foraminiferal $\delta^{13}C$ values (Bickert and Wefer, 1999). Hence, we measured the carbon isotopes of the foraminifera in our record and compare our data to that of other sites in the South Atlantic, which suggests that the values measured here represent a stable South Atlantic water mass signal. We do acknowledge that this integrated signal actually may represent different processes, which is actually one of the main points of discussion.

In the manuscript, we suggested and applied a 2.4 ‰ offset assuming a continuous presence of AAIW at the site (at lines 610-611), which is based on observing similar benthic foraminiferal $\delta^{13}C$ values compared to those previously reported by Curry and Oppo (2005) from the western Atlantic. They measured $\delta^{13}C$ of *Cibicidoides* (various species) from the South Atlantic along the Brazil margin spanning a depth range of 400 to 3000 m, which is affected by the same water mas (AAIW) as our core site (1375 meter water depth). They present an average glacial $\delta^{13}C$ value of 0.5 ‰ at a depth of 1500 m, which agrees well with the values measured here. Their results suggest a glacial-interglacial $\delta^{13}C$ difference of -0.1 ‰ whereas we observe a difference of -0.2 ‰. Although a difference of 0.1 ‰ may still be the result of seasonal water mass

instability (or many other processes), it does not affect our final conclusions. We will rephrase and clarify this in Section 5.2.

**5. Impact of Benguela Upwelling on atmospheric pCO₂.** The manuscript sort of kicks around this idea that changes in upwelling intensity and/or its carbon content might have altered atmospheric $pCO_2$. In the biological carbon pump/preformed nutrient content view of ocean $CO_2$ uptake/release (e.g., Sigman et al. 2010), though, whether or not the BUS was a $CO_2$ source or sink would have minimal impact on global atmospheric $CO_2$ This is because any excess upwelled nutrients in the region would then be advected to regions where they would be consumed. That is, even a high rate of local $CO_2$ outgassing at the core site due to upwelling > productivity would be offset by adjacent regions, where productivity > upwelling and $CO_2$ uptake would occur. Put another way, the only places where upwelling has the capacity to alter atmospheric $CO_2$ is in regions where that upwelling adds nutrients to newly formed deep water, either around Antarctica or in the high latitude Northern Hemisphere. In all other regions, local imbalances are evened out spatially.

**Response:** We agree with this comment, which is very much in line with what we are trying to argue. As this was apparently not completely clear we will rephrase our text accordingly. Results of our study suggest that even though an enhanced amount of $CO_2$ was likely stored at intermediate depth, this was not outgassed during the glacial due to enhanced productivity.

Minor comment (just one for now). Suggest changing the title to "Contrasts in the marine inorganic carbon chemistry of the Benguela Upwelling System since the Last Glacial Maximum". "Glacial-interglacial" implies that there are multiple data realizations of glacial intervals and interglacial intervals, so I found myself surprised when the data only went back to 27 ka.

**Response:** The title will be changed to "Contrasts in the marine inorganic carbon chemistry of the Benguela Upwelling System since the Last Glacial Maximum".

References:

Badger, M.P., Chalk, T.B., Foster, G.L., Bown, P.R., Gibbs, S.J., Sexton, P.F., Schmidt, D.N., Pälike, H., Mackensen, A. and Pancost, R.D., 2019. Insensitivity of alkenone carbon isotopes to atmospheric CO2 at low to moderate CO2 levels. *Climate of the Past*, *15*(2), 539-554.

Cenozoic CO2 Proxy Integration Project (CenCO2PIP) Consortium*†, 2023. Toward a Cenozoic history of atmospheric CO₂. *Science*, *382*(6675), eadi5177.

Farmer, J. R., Hertzberg, J. E., Cardinal, D., Fietz, S., Hendry, K., Jaccard, S. L., et al. (2021). Assessment of C, N, and Si isotopes as tracers of past ocean nutrient and carbon cycling. *Global Biogeochemical Cycles*, 35, e2020GB006775.

Henehan, M.J., Rae, J.W., Foster, G.L., Erez, J., Prentice, K.C., Kucera, M., Bostock, H.C., Martínez-Botí, M.A., Milton, J.A., Wilson, P.A. and Marshall, B.J., 2013. Calibration of the boron isotope proxy in the planktonic foraminifera Globigerinoides ruber for use in palaeo-CO2 reconstruction. *Earth and Planetary Science Letters*, *364*, 111-122.

Henehan, M.J., Foster, G.L., Bostock, H.C., Greenop, R., Marshall, B.J. and Wilson, P.A., 2016. A new boron isotope-pH calibration for Orbulina universa, with implications for understanding and accounting for 'vital effects'. *Earth and Planetary Science Letters*, *454*, 282-292.

Hönisch, B. and Hemming, N.G., 2004. Ground-truthing the boron isotope-paleo-pH proxy in planktonic foraminifera shells: Partial dissolution and shell size effects. *Paleoceanography*, *19*(4).

Martínez-Botí, M.A., Marino, G., Foster, G.L., Ziveri, P., Henehan, M.J., Rae, J.W., Mortyn, P.G. and Vance, D., 2015. Boron isotope evidence for oceanic carbon dioxide leakage during the last deglaciation. *Nature*, *518*(7538), 219-222.

Phelps, S.R., Hennon, G.M., Dyhrman, S.T., Hernández Limón, M.D., Williamson, O.M. and Polissar, P.J., 2021. Carbon isotope fractionation in Noelaerhabdaceae algae in culture and a critical evaluation of the alkenone paleobarometer. *Geochemistry, Geophysics, Geosystems*, *22*(7), e2021GC009657.

Rae, J.W., Foster, G.L., Schmidt, D.N. and Elliott, T., 2011. Boron isotopes and B/Ca in benthic foraminifera: Proxies for the deep ocean carbonate system. *Earth and Planetary Science Letters*, *302*(3-4), 403-413.

Sigman, D.M., Hain, M.P. and Haug, G.H., 2010. The polar ocean and glacial cycles in atmospheric CO2 concentration. *Nature*, *466*(7302), 47-55.

References

Badger, M. P. S.: Alkenone isotopes show evidence of active carbon concentrating mechanisms in coccolithophores as aqueous carbon dioxide concentrations fall below 7 μmol $L^{-1}$, *Biogeosciences*, 18, 1149-1160, https://doi.org/10.5194/bg-18-1149-2021, 2021.

Bickert, T. and Wefer, G.: South Atlantic and benthic foraminifer $\delta^{13}C$ deviations: implications for reconstructing the Late Quaternary deep-water circulation, *Deep Sea Research Part II: Topical Studies in Oceanography*, 46, 437-452, https://doi.org/https://doi.org/10.1016/S0967-0645(98)00098-8, 1999.

Curry, W. B. and Oppo, D. W.: Glacial water mass geometry and the distribution of $\delta^{13}C$ of $\Sigma CO_2$ in the western Atlantic Ocean, *Paleoceanography*, 20, https://doi.org/10.1029/2004PA001021, 2005.

Goudet, M. M., Orr, D. J., Melkonian, M., Müller, K. H., Meyer, M. T., Carmo-Silva, E., and Griffiths, H.: Rubisco and carbon-concentrating mechanism co-evolution across chlorophyte and streptophyte green algae, *New Phytol.*, 227, 810-823, https://doi.org/10.1111/nph.16577, 2020.

Hönisch, B., Allen, K. A., Russell, A. D., Eggins, S. M., Bijma, J., Spero, H. J., Lea, D. W., and Yu, J.: Planktic foraminifers as recorders of seawater Ba/Ca, Mar. *Micropaleontol.*, 79, 52-57, https://doi.org/10.1016/j.marmicro.2011.01.003, 2011.

Jouzel, J., Masson-Delmotte, V., Cattani, O., Dreyfus, G., Falourd, S., Hoffmann, G., Minster, B., Nouet, J., Barnola, J. M., Chappellaz, J., Fischer, H., Gallet, J. C., Johnsen, S., Leuenberger, M., Loulergue, L., Luethi, D., Oerter, H., Parrenin, F., Raisbeck, G., Raynaud, D., Schilt, A., Schwander, J., Selmo, E., Souchez, R., Spahni, R., Stauffer, B., Steffensen, J. P., Stenni, B., Stocker, T. F., Tison, J. L., Werner, M., and Wolff, E. W.: Orbital and Millennial Antarctic Climate Variability over the Past 800,000 Years, Science, 317, 793-796, https://doi.org/10.1126/science.1141038, 2007.

Lauvset, S. K., Lange, N., Tanhua, T., Bittig, H. C., Olsen, A., Kozyr, A., Álvarez, M., Azetsu-Scott, K., Brown, P. J., Carter, B. R., Cotrim da Cunha, L., Hoppema, M., Humphreys, M. P., Ishii, M., Jeansson, E., Murata, A., Müller, J. D., Perez, F. F., Schirnick, C., Steinfeldt, R., Suzuki, T., Ulfsbo, A., Velo, A., Woosley, R. J., and Key, R.: The annual update GLODAPv2.2023: the global interior ocean biogeochemical data product, *Earth Syst. Sci. Data Discuss.*, 2024, 1-32, https://doi.org/10.5194/essd-2023-468, 2024.

North Greenland Ice Core Project members: High-resolution record of Northern Hemisphere climate extending into the last interglacial period, *Nature*, 431, 147-151, https://doi.org/10.1038/nature02805, 2004.

Petit, J. R., Jouzel, J., Raynaud, D., Barkov, N. I., Barnola, J. M., Basile, I., Bender, M., Chappellaz, J., Davis, M., Delaygue, G., Delmotte, M., Kotlyakov, V. M., Legrand, M., Lipenkov, V. Y., Lorius, C., PÉpin, L., Ritz, C., Saltzman, E., and Stievenard, M.: Climate and atmospheric history of the past 420,000 years from the Vostok ice core, Antarctica, *Nature*, 399, 429-436, https://doi.org/10.1038/20859, 1999.

Reinfelder, J. R.: Carbon Concentrating Mechanisms in Eukaryotic Marine Phytoplankton, *Ann. Rev. Mar. Sci.*, 3, 291-315, https://doi.org/10.1146/annurev-marine-120709-142720, 2011.

Riebesell, U., Wolf-Gladrow, D. A., and Smetacek, V.: Carbon dioxide limitation of marine phytoplankton growth rates, *Nature*, 361, 249-251, https://doi.org/10.1038/361249a0, 1993.

Salt, L. A., van Heuven, S. M. A. C., Claus, M. E., Jones, E. M., and de Baar, H. J. W.: Rapid acidification of mode and intermediate waters in the southwestern Atlantic Ocean, *Biogeosciences*, 12, 1387-1401, https://doi.org/10.5194/bg-12-1387-2015, 2015.

Stuut, J.-B. W., Prins, M. A., Schneider, R. R., Weltje, G. J., Jansen, J. H. F., and Postma, G.: A 300-kyr record of aridity and wind strength in southwestern Africa: inferences from grain-size distributions of sediments on Walvis Ridge, SE Atlantic, *Mar. Geol.*, 180, 221-233, https://doi.org/10.1016/S0025-3227(01)00215-8, 2002.

---

## Author Comment (AC2)

Karancz et al. reconstruct past ocean conditions for the past ~27 cal ky BP using a suite of geochemical proxies from a box and piston core collected from the south flank of Walvis Ridge in the South Atlantic. Multiple techniques are applied to reconstruct seawater temperature and pCO2, representing a major analytical effort by the author team.

There are some issues with the manuscript that should be addressed prior to publication, described in more detail below.

**Response:** Hereby we would like to thank the reviewer for the insightful comments which helped to clarify and improve the manuscript. Below we provide a detailed response to each of the comments.

1. There is an age reversal in the box core that complicates interpretation of the Holocene portions of all the geochemical records (~5-10 cal ky BP). The authors do not discuss the age reversal, keeping it in the age model (Fig. 3a), and suggesting a sedimentation rate of 0.01 cm/yr, or "Alternatively, the upper 10 cm bsf have constant ages due to bioturbation." This is not a clear or satisfying discussion of the observed radiocarbon ages in the box core. On the figure, there is also a note "winnowing (?)" with no discussion of winnowing in the main text. In addition, if the aim of the study is to gain insight into millennial scale events using the geochemical records from this core, then it could be worth investing in tighter age control through more radiocarbon dates per depth interval (e.g., at least 1 age control point per 2 ky; Guilderson et al., 2021). At a minimum, the box core's chronology is uncertain and the basis for splicing the box and piston core has not been clearly described (as noted by reviewer 1 in the previous comment).

**Response:** We agree with the comments on core stratigraphy from both Reviewer #1 and #2 and changed our age model strategy accordingly. We now use measured brightness (l*) of detailed line-scan pictures of sediment cores BC6 and PC8, added to the Supplementary material, to provide a stronger underpinning for the age model and tying the two cores together. For this figure and an additional figure showing the overlap of BC6 and PC8 based on Log(Ca/Ti) from XRF-scanning, please see our response to the first review comment from Jesse Farmer.

The upper 11 cm of box core BC6 covers 0.55 ka, with somewhat elevated Ca/Al, Ti/Al and Si/Al ratios (generated by XRF core-scanning). Enrichment of elements that preferably occur in coarse fractions and in heavy minerals is likely the result of the fine fraction being removed by winnowing (e.g., Karageorgis et al., 2005). Hence, we here may interpret the loss of the last 4.8 ka from the sedimentary record as an increase of winnowing activity. We will discuss the possibility of winnowing in the main text and include the figures with the XRF profiles in the Supplementary material.

[Figure]

*Figure S2. Log(Ca/Al), Log(Ti/Al), and Log(Si/Al) ratios (XRF-scanning element intensities) of 64PE450-BC6 plotted over the last 10 ka BP.*

2. After going through the major effort of generating several complementary geochemical records, it is surprising that the authors do not synthesize the data to the full extent possible. The most prominent missed opportunity is that, if I understand correctly, the authors apply a uniform temperature correction across the whole G. bulloides carbon isotope record, from ~5 to 27 cal ky BP. Why not use the available Mg/Ca derived temperatures and the established temperature dependence of d13C for G. bulloides to get the most accurate values possible? There is a temperature-corrected curve in the supplement (Fig. S3), but it looks the same as the gray curve in main text Fig. 6C, which was created by applying a constant offset, so it appears not to have been created using the variable, reconstructed temperature estimates (it would be nice if the Fig. S3 caption could be updated to clarify that). There is also a carbonate-ion corrected curve in the supplement (Fig. S3), but it is not clear how the paleo carbonate ion values were determined or why this correction was not also applied to the record that was ultimately compared with benthic isotopes to assess changes in the soft tissue pump in the main text.

**Response:** We thank the reviewer for this comment and will change the graphs accordingly. The correction made in the original manuscript, using a constant value of 2.4 ‰, was already based on temperature corrections derived from Mg/Ca values, but not yet calculated for individual points. Hence, the values shown on Figure 6c are similar to those on Figure S3, but trends may differ somewhat. Temperature does not change much during the investigated interval (14.7 – 16.7 °C), corresponding to a $\delta^{13}C$ offset between 2.4 – 2.6 ‰. As the contribution of carbonate ion (and potentially other) effect to these offsets is uncertain, we applied a constant correction value which agrees well with the modern offset to the $\delta^{13}C$ values of DIC.

The carbonate ion corrected curve is based on $[CO_3^{2-}]$ calculated from S/Mg values in the original manuscript. However, based on a comment from Reviewer #1, this approach will be removed and subsequently we will use a correction based on $[CO_3^{2-}]$ derived from pH and total alkalinity.

Although the biological carbon pump cannot be quantitatively determined within this study, the trend in the observed planktonic $\delta^{13}C$ record is robust. Independently of the applied corrections, the difference between planktonic and benthic $\delta^{13}C$ values remains larger during the LGM compared to the Holocene. The curve showing the correction with constant value will be moved to the Supplementary figure, and Figure 6 in the main text will be modified to display combined correction for both temperature and $[CO_3^{2-}]$. Captions for both Figure S3 and Figure 6 will be adjusted accordingly.

[Figure]

*Adjusted Figure S3. Measured $\delta^{13}C$ values of G. bulloides (dark blue diamonds), and C. wuellerstorfi (red diamonds), and $\delta^{13}C$ values of G. bulloides corrected for temperature (purple triangles; Bemis et al., 2000), $[CO_3^{2-}]$ (green triangles; Bijma et al., 1999), and for a constant offset of 2.4 ‰ (grey diamonds, corresponding to the modern offset to $\delta^{13}C$ of DIC) plotted over the past 27 ka BP. Green diamonds display $\delta^{13}C$ values corrected for both temperature (derived from Mg/Ca; Bemis et al., 2000) and $[CO_3^{2-}]$ (derived from pH and TA; Bijma et al., 1999).*

[Figure]

*Adjusted Figure 6: Reconstructed sea surface temperatures (SST) based on a) the alkenone unsaturation index, $U^K_{37}$, and b) foraminiferal Mg/Ca, c) $\delta^{13}C$ analysed in benthic (C. wuellerstorfi) and planktonic (G. bulloides) foraminifera with corrected values, d) $\delta^{18}O$ of benthic (C. wuellerstorfi) and planktonic (G. bulloides) foraminifera, and e) $\delta^{18}O$ ice core record from EPICA-Dome C (EDC; Jouzel et al., 2007) and North Greenland Ice Core Project (NGRIP; North Greenland Ice Core Project members, 2004) shown for the past 27 ka BP. Corrected $\delta^{13}C$ values of G. bulloides marked with green diamonds are based on temperature (derived from Mg/Ca; Bemis et al., 2000) and $[CO_3^{2-}]$ values (derived from pH and TA; Bijma et al., 1999). Modern day SST at core site 64PE450-BC6-PC8 is approximately 20.7 °C (GLODAPv2023; Lauvset et al., 2024; Santana-Casiano et al., 540 2009). Grey shaded areas mark climate events as labelled on the uppermost panel. Blue shaded area in panel b) indicates the error propagated from temperature calibration uncertainty and ± 1σ standard deviation of the duplicate measurement of the samples. Analysis of the stable isotopes (panel c and d) provided an error smaller than the symbols shown on the figure. Arrows in panel c) indicate the direction of the correction on the planktonic foraminiferal $\delta^{13}C$.*

3. In the discussion, the use of "intermediate" is confusing, sometimes being applied to G. bulloides records (which can live anywhere from the shallow surface down to a couple hundred meters) and sometimes being applied to benthic foraminiferal records. The cores were collected at 1375m water depth, which lies within the modern extent of AAIW in the region, so discussing the benthic foraminiferal records in the context of AAIW is reasonable and is also supported by prior studies. However, use of "intermediate" to describe the G. bulloides records is misleading, since there is no evidence that this species inhabits AAIW. Instead, G. bulloides tends to live very shallowly during the active upwelling season and extend more deeply in other seasons (e.g., Peeters & Brummer, 2002). Karancz et al. suggest, based on Mg/Ca-derived temperatures, that their Holocene bulloides are living at ~100-150m (~line 515), and this is shallower than the uppermost extent of AAIW. I'd strongly recommend using another word (like "subsurface") to describe the G. bulloides record, and reserve use of "intermediate" for AAIW alone.

**Response:** Thank you for pointing this out as the word "intermediate" has been indeed used in two contexts which may lead to confusion. We will rephrase the relevant sentences and use subsurface to describe data related to *G. bulloides*.

4. Considering the wide range of possible habitat depths (and seasonal variability) of G. bulloides, comparing G. bulloides d11B-derived pCO2 and alkenone-based pCO2 is not a robust approach for reconstructing an intermediate-to-surface pCO2 gradient.

**Response:** We agree and the differences observed here will have a minimum offset and cannot be interpreted quantitively. Still, *G. bulloides* living in the subsurface and migrating between approximately 50 to 400 m, represents a larger average water depth then the alkenone record (~upper 50 m). The depth difference between *G. bulloides* and alkenone-producers may vary seasonally (and also on longer time scales) but *G. bulloides* still represents carbon system conditions closer to those of the upwelled intermediate waters. We acknowledge that the water depth of $pCO_2$ values represented by *G. bulloides* - $\delta^{11}B$ are most likely not constant and this will be added in the Discussion.

5. The discussion would be strengthened by providing specific references and paleorecords when comparing their results with prior work. There are multiple instances (line 530 "changes known for the Southern Hemisphere"; line 500 "… general Southern Hemisphere temperature record") where paleoclimate patterns are alluded to but not cited or described specifically. Changes were not uniform across the SH during the last deglaciation and therefore it is essential that the authors are very explicit about what they are referring to.

**Response:** The discussion will be modified to include both Southern and Northern Hemisphere examples with specific references as the individual climate events also show similarities to the trends observed in Northern Hemisphere records.

lines 500 – 505: references for Southern Hemisphere showing a warming signal during the Younger Dryas e.g., Rühlemann et al., 1999; Singer et al., 1998; Bennett et al., 2000; Jouzel et al., 2007

lines 530 – 534: references for climate/temperature changes known for the Southern Hemisphere over the past 27 ka BP e.g., Johnsen et al., 1972; Blunier and Brook, 2001; Sugden et al., 2005; Jouzel et al., 2007; Brook and Buizert, 2018.

The data presented here represent an enormous amount of work, and they do have the potential to make a useful contribution to our understanding of paleoceanography in the South Atlantic. I hope that these comments are useful to the authors.

References:

Guilderson, T. P., Allen, K., Landers, J. P., Ettwein, V. J., & Cook, M. S. (2021). Can We Better Constrain the Timing of GNAIW/UNADW Variability in the Western Equatorial Atlantic and Its Relationship to Climate Change During the Last Deglaciation? *Paleoceanography and Paleoclimatology*, *36*(8). https://doi.org/10.1029/2020pa004187

Peeters, F. J. C., & Brummer, G.-J. A. (2002). The seasonal and vertical distribution of living planktic foraminifera in the NW Arabian Sea. *Geological Society, London, Special Publications*, *195*(1), 463–497. https://doi.org/10.1144/gsl.sp.2002.195.01.26

References:

Bennett, K. D., Haberle, S. G., and Lumley, S. H.: The Last Glacial-Holocene Transition in Southern Chile, Science, 290, 325-328, https://doi.org/10.1126/science.290.5490.325, 2000.

Blunier, T. and Brook, E. J.: Timing of Millennial-Scale Climate Change in Antarctica and Greenland During the Last Glacial Period, Science, 291, 109-112, https://doi.org/10.1126/science.291.5501.109, 2001.

Brook, E. J. and Buizert, C.: Antarctic and global climate history viewed from ice cores, Nature, 558, 200-208, https://doi.org/10.1038/s41586-018-0172-5, 2018.

Johnsen, S. J., Dansgaard, W., Clausen, H. B., and Langway, C. C.: Oxygen Isotope Profiles through the Antarctic and Greenland Ice Sheets, Nature, 235, 429-434, https://doi.org/10.1038/235429a0, 1972.

Jouzel, J., Masson-Delmotte, V., Cattani, O., Dreyfus, G., Falourd, S., Hoffmann, G., Minster, B., Nouet, J., Barnola, J. M., Chappellaz, J., Fischer, H., Gallet, J. C., Johnsen, S., Leuenberger, M., Loulergue, L., Luethi, D., Oerter, H., Parrenin, F., Raisbeck, G., Raynaud, D., Schilt, A., Schwander, J., Selmo, E., Souchez, R., Spahni, R., Stauffer, B., Steffensen, J. P., Stenni, B., Stocker, T. F., Tison, J. L., Werner, M., and Wolff, E. W.: Orbital and Millennial Antarctic Climate Variability over the Past 800,000 Years, Science, 317, 793-796, https://doi.org/10.1126/science.1141038, 2007.

Karageorgis, A. P., Kaberi, H., Price, N. B., Muir, G. K. P., Pates, J. M., and Lykousis, V.: Chemical composition of short sediment cores from Thermaikos Gulf (Eastern Mediterranean): Sediment accumulation rates, trawling and winnowing effects, *Continental Shelf Research*, 25, 2456-2475, https://doi.org/10.1016/j.csr.2005.08.006, 2005.

Rühlemann, C., Mulitza, S., Müller, P. J., Wefer, G., and Zahn, R.: Warming of the tropical Atlantic Ocean and slowdown of thermohaline circulation during the last deglaciation, Nature, 402, 511-514, https://doi.org/10.1038/990069, 1999.

Singer, C., Shulmeister, J., and McLea, B.: Evidence Against a Significant Younger Dryas Cooling Event in New Zealand, Science, 281, 812-814, https://doi.org/10.1126/science.281.5378.812, 1998.

Sugden, D. E., Bentley, M. J., Fogwill, C. J., Hulton, N. R. j., McCulloch, R. D., and Purves, R. S.: Late-glacial glacier events in southernmost south america: a blend of 'northern' and 'southern' hemispheric climatic signals?, Geografiska Annaler: Series A, Physical Geography, 87, 273-288, https://doi.org/10.1111/j.0435-3676.2005.00259.x, 2005.

---

## Referee Report (RR1)

Review of Karancz et al., "Glacial-interglacial contrasts in the marine inorganic carbon chemistry of the Benguela Upwelling System," revised for *Climate of the Past*, by Jesse Farmer

I greatly appreciated the authors' thoughtful and detailed responses, and largely agree with their revisions. At the same time, a closer read of the manuscript illuminates multiple areas that need refinement before publication. My recommendation is another round of major revisions here, but I'm hopeful that the revised manuscript will be publishable in *Climate of the Past*.

Major comments.

(1) **Structure of Introduction section.** The current introduction section (L32-105) is rather challenging to follow as it covers massive scale differences, from global upwelling regions (L32-40) to the importance of alkenone physiology to pCO2 reconstructions (L85). To help the reader, I'd recommend keeping the current L32-61, then L96-105 (plus "Here" sentence starting on L74) as the introduction. The details of the proxies themselves (L63-94) could be a separate subsection titled "Proxy Interpretation" in the introduction, as a separate section after "Oceanographic Setting" (better), or perhaps as a subsection of the Methods.

(2) **Discussion Section 5.1.** Within this section, I find that the text does not accurately reflect the data, especially the poor correspondence between local temperature proxies (Mg/Ca, Uk'37, d18O; e.g., L496-541), and their poor correspondence to high latitude temperature indicators (e.g., L488-494). It is completely fine that different temperature reconstructions do not agree; this happens all the time in paleoceanography. But the authors must be honest in their description. Moreover, when records do not correlate, a mechanistic discussion is not warranted because it is difficult to gain any insight on the underlying mechansims (e.g., L509-516).

Overall, I believe the authors should greatly shorten this section, noting that some proxies (Uk'37, d18O) appear to show a "classic" deglaciation pattern, while other proxies (Mg/Ca) do not. I think the major takeaway of this section should be that the dissimilarity within the different paleotemperature proxies at the site may reflect changing and complex upper water column temperature structure over the last 27,000 years, while the dissimilarity between these proxies and high latitude temperature proxies probably reflects changing and complex interactions between temperature at the core location and those in the high northern and southern latitudes.

(3) **Section 5.4 needs to be overhauled.** Some ideas:
   A. The authors should focus first on their primary observation that is shown in Figure 8. After that, they can speculate that G. bulloides might more faithfully reflect the properties of AAIW due to its subsurface depth.

B. Please also draw an arrow for atmospheric pCO2 on Fig 8.

C. After this, they can discuss potential local influences on pCO2 gradients (L749-755) and rule these out.

D. Following ruling out the local processes, they can conclude by suggesting their data are evidence for greater CO2 storage in mid-depth waters during the LGM, as has been previously suggested. They do not need to get into mechanistic explanations (e.g., local vs. remote iron fertilization) because their data do not speak to the underlying mechanisms. (Unless they also wish to add an iron flux record to this!)

**Line-by-line comments/edits.**

Abstract L18-19. Suggested rephrase: "but also on the efficiency of the biological carbon pump, which constrains the drawdown of atmospheric $CO_2$ in the surface waters."

L59-61. I think it would be best to briefly summarize the caveats to this 1-dimensional approach here. Namely, ocean circulation is constantly working to "erase" the surface to deep gradients, while also importing waters with different d13C signatures due to the integrated histories of air-sea exchange and production/remineralization within those watermasses.

L71. Suggest adding paragraph break at "Here", and removing the paragraph break on L77.

L85-86. Delete comma before "that"

L121. Grammar – change its' to its (delete apostrophe); see correct usage on L133.

L124-125. Change "Corilois-force" to "Corilois force"

L129. Delete high, as the word "productive" implies high productivity

L145. Suggest change to "…year-round upwelling of varying intensity due to…"

L147. Suggest clarification to "Predominantly, the surface waters within the BUS act as a CO2 source…"

L165. Here you should move up the paragraph presenting the L* data, currently on L414-421. When you move this paragraph up, please delete the clause "which together comprise a near-continuous time interval from ~5 to 27 ka BP" as you will not have discussed ages

yet at this point in the methods. Note also that light reflectance is typically signified as L* (capitalized L); this should be changed here and in Figure S2.

L164. Delete space before end parenthesis

L170. Change to "relatively high abundance"

L173-184. This is a step in the right direction, but the revised version is misleading. I interpreted this new text to mean that the authors have picked from a narrow size fraction for C and O isotopes, because ontogeny matters for these isotope systems, but they picked over a broad size fraction for B isotopes, because ontogeny doesn't matter for symbiont-barren foraminifera in the B isotope system. The problem is that we do not know whether this is true; the variation in the B isotopic composition of *G. bulloides* size fractions has not been tested (pending publication of the Buisson et al. results). There are reasons to suspect that different *G. bulloides* sizes may be living in different depths in the water column (e.g., Jonkers et al., 2013; Osborne et al., 2016; but cf. Metcalfe et al., 2015), which could influence what environmental pH they record. As a result, there is an uncertainty here that the authors must acknowledge even if they cannot quantify it.

To address this, I'd recommend the following roadmap:
1. (As the authors currently do) acknowledge that the larger sample size requirements of 14C, d11B, and El/Ca measurements required picking G. bulloides from the 150-425 µm size fraction
2. (Also as the authors do) note that there are observed size fraction d11B differences in symbiont-bearing planktonic foraminifera.
3. Note that while symbiosis should not be a concern for G. bulloides, uncertainty in the environmental conditions reflected by different size fractions could lead to biasing of the d11B and El/Ca results. In addition, previous studies looking at G. bulloides d11B have worked over narrow size fractions (Martínez-Botí et al., 2015; Raitzsch et al., 2018).
4. They can then cite Buisson et al. (in review) here to discount this effect, and/or note that future work would be need to evaluate any size fraction-specific influences.
5. It is fine to say that, for the sake of this manuscript, you assume that the d11B and El/Ca results in G. bulloides reflect average conditions.

Jonkers, L., S. van Heuven, R. Zahn, and F. J. C. Peeters (2013), Seasonal patterns of shell flux, $\delta^{18}O$ and $\delta^{13}C$ of small and large *N. pachyderma* (s) and *G. bulloides* in the subpolar North Atlantic, *Paleoceanography*, 28, 164–174, doi:10.1002/palo.20018.

Metcalfe, B., W. Feldmeijer, M. de Vringer-Picon, G. J. Brummer, G.J., F. J. C. Peeters, and G. M. Ganssen (2015), Late Pleistocene glacial–interglacial shell-size–isotope variability in planktonic foraminifera as a function of local hydrography, *Biogeosciences*, *12*(15), 4781-4807.

Osborne, E. B., R. C. Thunell, B. J. Marshall, J. A. Holm, E. J. Tappa, C. Benitez-Nelson, W.-J. Cai, and B. Chen (2016), Calcification of the planktonic foraminifera *Globigerina bulloides* and carbonate ion concentration: Results from the Santa Barbara Basin, *Paleoceanography*, 31, 1083–1102, doi:10.1002/2016PA002933.

L191. Change "sonification" to "sonication"

L193. Change to "removal of"

L209. Change to "exetainer"

L226-227. Please specify whether the reported d13C/d18O values are for the NFHS-1 standard; it is unclear as written.

L286-297. As you already had to calculate past salinity for the carbonate chemistry calculation (L294-295), you should evaluate the effect of changing salinity on Mg/Ca-dervied SST. With very little additional work, you can quantify the assertion that the effect of salinity on temperature is "relatively minor" (L290).

L312-314. In order to solve for CO2, you need to specify a second variable in the carbonate system. Which variable did you specify, and how did you estimate its value? See e.g. L668 far later down – there needs to be detailed methods to define your approach.

Section 4.1. Please report sedimentation rates in cm kyr$^{-1}$ instead of cm yr$^{-1}$.

L443. Most of the rapid changes in d18O predate the deglaciation; e.g. the variance between 25 and 20 ka occurs during the LGM (dated to 26.5 – 19 ka, e.g. Clark et al., 2009, cited in manuscript).

L455, L469 and throughout. Either "last 6 kyr" or "since 6 ka BP". Kyr is a block of time (6,000 years), ka BP is a specific date (6,000 years ago).

L500-504. I find this meltwater hypothesis difficult to fathom – at face value, one would expect that increased meltwater would reduce the formation rate of AAIW due to buoyancy gain (see, e.g., Starr et al., 2021), and thus the low-d18O signature of meltwater would not be upwelled in the Benguela region. Has this meltwater hypothesis been suggested anywhere previously? I'd definitely recommend adding references and/or expressing greater uncertainty in this explanation.

Starr, A., Hall, I.R., Barker, S., Rackow, T., Zhang, X., Hemming, S.R., van der Lubbe, H.J.L., Knorr, G., Berke, M.A., Bigg, G.R. and Cartagena-Sierra, A., 2021. Antarctic icebergs reorganize ocean circulation during Pleistocene glacials. *Nature*, *589*(7841), 236-241.

L496-540. Why is the discussion of Mg/Ca left separately?

L541. To emphasize the point above, the Mg/Ca reconstruction from G. bulloides is not "generally in line with the [alkenone] and d18O record" and does not "confirm overall trends". In fact, the G. bulloides Mg/Ca-temperature only looks like alkenone and d18O records during the warming between about 23 and 16 ka. Outside of that, this is a completely different record.

L542-558. This is a good discussion of potential depth offsets and should be kept in a revised and shortened Section 5.1.

L595. Change "a" to "their"

L609-611. It is unclear what the stated d13C values here (0.3-1.0‰ and 2.4-2.6‰) refer to – are they the magnitudes of d13C corrections per process? The sensitivity of d13C to each process? Please specify or remove these values.

L619-626. I believe this is out of order – on L619 the sentences "We here applied... The offset of..." should come first, then the results from the correction application should follow: "Still, when corrections..."

L628-646. For comparing to other sites putatitvely in AAIW, there are more recent publications than Curry and Oppo (2005). Please compare your results against these.

Lacerra, M., Lund, D.C., Gebbie, G., Oppo, D.W., Yu, J., Schmittner, A. and Umling, N.E. (2019), Less Remineralized Carbon in the Intermediate-Depth South Atlantic During Heinrich Stadial 1. Paleoceanography and Paleoclimatology, 34: 1218-1233. https://doi.org/10.1029/2018PA003537

Umling, N.E., Oppo, D.W., Chen, P., Yu, J., Liu, Z., Yan, M., Gebbie, G., Lund, D.C., Pietro, K.R., Jin, Z.D., Huang, K.-.-F., Costa, K.B. and Toledo, F.A.L. (2019), Atlantic Circulation and Ice Sheet Influences on Upper South Atlantic Temperatures During the Last Deglaciation. Paleoceanography and Paleoclimatology, 34: 990-1005. https://doi.org/10.1029/2019PA003558

L643. Delete repeated "values"

L667. See point above about specifying your approach for calculating $pCO_2$ from d11B in the methods.

L668-670. Please delete phrase "which is likely related to AMOC intensity" as there is no mechanistic connection presented as to why reduced AMOC should lead to pCO2 equilibrium calculated from d11B. (Alternatively, you can bring the reader through the mechanistic connections that would create this expectation).

L680. Delete "remarkably" – as you point out in the next sentence, there's a ~65 ppm offset during about half of the Holocene. That is 2/3rds of the glacial-interglacial difference in CO2, so it is certainly a significant offset.

L682. "this remained more or less in equilibrium" – what is "this"??

L695-704. The argument presented here is that using Ba/Ca as a constraint on phosphate concentration provides better estimation of the *b* factor than assuming constant phosphate. But the authors have not presented data to back this case up. How does calculated pCO2 compare when using Ba/Ca vs. assuming constant phosphate? This comparison must be presented to justify the conclusion given on L699-704.

L788-789. Again, delete "remarkably"; perhaps say "while surface values of pCO2 reconstructed from d13C of alkenones generally track atmospheric pCO2"